# Structural and functional properties of a plant NRAMP-related aluminum transporter

**Karthik Ramanadane, Márton Liziczai, Dragana Markovic, Monique S Straub, Gian T Rosalen, Anto Udovcic, Raimund Dutzler*, Cristina Manatschal***

Department of Biochemistry, University of Zurich, Zurich, Switzerland

**Abstract** The transport of transition metal ions by members of the SLC11/NRAMP family constitutes a ubiquitous mechanism for the uptake of $Fe^{2+}$ and $Mn^{2+}$ across all kingdoms of life. Despite the strong conservation of the family, two of its branches have evolved a distinct substrate preference with one mediating $Mg^{2+}$ uptake in prokaryotes and another the transport of $Al^{3+}$ into plant cells. Our previous work on the SLC11 transporter from *Eggerthella lenta* revealed the basis for its $Mg^{2+}$ selectivity (Ramanadane et al., 2022). Here, we have addressed the structural and functional properties of a putative $Al^{3+}$ transporter from *Setaria italica*. We show that the protein transports diverse divalent metal ions and binds the trivalent ions $Al^{3+}$ and $Ga^{3+}$, which are both presumable substrates. Its cryo-electron microscopy (cryo-EM) structure displays an occluded conformation that is closer to an inward- than an outward-facing state, with a binding site that is remodeled to accommodate the increased charge density of its transported substrate.

## Editor's evaluation

This study uses cryo-EM and functional assays to characterize a new subtype of transporters from the NRAMP family that protect plants against aluminum toxicity. Evidence of divalent metal ion transport and the structure (obtained without added metal ions) are convincing. Although technical challenges inherent to the structural and functional characterization of eukaryotic membrane proteins prevent a complete molecular description of $Al^{3+}$ binding and transport, the work nonetheless provides new and valuable insight into the diversity within the NRAMP superfamily of transporters, and will be of interest to structural biologists and biophysicists studying NRAMP transporters.

**\*For correspondence:**
dutzler@bioc.uzh.ch (RD);
c.manatschal@bioc.uzh.ch (CM)

**Competing interest:** The authors declare that no competing interests exist.

## Introduction

Owing to their unique properties to form coordinative interactions with elements containing free electron pairs and their ability to readily change their oxidation states, $Fe^{2+}$ and $Mn^{2+}$ are important trace elements in biology. The uptake of both ions into cells requires an elaborate mechanism that allows their selection from a large background of alkaline earth metal ions and their efficient concentration within cells. This formidable task is performed by members of the SLC11 family, which constitute a conserved family of secondary active transporters that facilitate the cotransport of transition metal ions together with protons acting as energy source (**Courville et al., 2006**; **Gunshin et al., 1997**; **Montalbetti et al., 2013**). To be able to do this across all kingdoms of life, the common molecular features conferring ion selectivity and proton coupling are strongly conserved within different branches of the family (**Bozzi and Gaudet, 2021**). However, despite this strong conservation, there are two distinct clades of the family, which have evolved a different substrate selectivity. These comprise the large group of

NRAMP-related $Mg^{2+}$ transporters (NRMTs) and a smaller group of NRAMP-related $Al^{3+}$ transporters (NRATs) (*Chauhan et al., 2021*; *Shin et al., 2014*; *Xia et al., 2010*). In a recent study, we have shown how NRMTs have adapted their molecular architecture to facilitate $Mg^{2+}$ uptake, by characterizing a transporter from the bacterium *Eggerthella lenta* termed EleNRMT (*Ramanadane et al., 2022*). Due to its small ionic radius, the stripping of the interacting solvent is particularly costly for $Mg^{2+}$, and its transport by EleNRMT thus likely proceeds as hydrated ion retaining most of its first coordination shell. Besides $Mg^{2+}$, the protein also transports $Mn^{2+}$ but not $Ca^{2+}$. In NRMTs, several of the residues coordinating ions in transition metal transporters of the family are exchanged, including a conserved methionine, which forms coordinative interactions with transition metals while preventing the binding of alkaline earth metal ions (*Bozzi et al., 2016*). The reduction of the side chain volume of interacting residues in combination with small rearrangements of the backbone have increased the volume of the binding site to accommodate the hydrated ion (*Ramanadane et al., 2022*). Unlike in proton-coupled family members, transport in EleNRMT is passive with residues that have been assigned to participate in $H^+$ transport being replaced by hydrophobic side chains (*Ramanadane et al., 2022*).

Besides the prokaryotic NRMTs, which constitute a large branch of the family that is distantly related to transition metal ion transporters of the same family, the NRATs form a small clade in plants, whose evolutionary relationship to classical SLC11 transporters is much closer (*Ramanadane et al., 2022*). These proteins have evolved to combat $Al^{3+}$ toxicity in soil (*Chauhan et al., 2021*). As strong Lewis acid, this trivalent cation strongly interacts with membranes to compromise their integrity. As one of the mechanisms to neutralize the adverse effect of $Al^{3+}$, NRATs were proposed to catalyze its uptake into the cytoplasm of root cells acting in concert with other proteins, which mediate the further transport into vacuoles where it is neutralized by the complexation with dicarboxylates (*Xia et al., 2010*).

To clarify the mechanism by which NRATs transport trivalent cations, we have here studied the function and structure of a transporter from *Setaria italica* (termed SiNRAT). We show that this protein broadly transports divalent transition metal ions by a mechanism that is not coupled to protons. Although, for technical reasons, transport of trivalent cations could not directly be assayed, we provide evidence for their specific interaction with the binding site, thus defining them as likely substrates. The cryo-EM structure of SiNRAT displays a protein that shares the hallmarks of the SLC11 family. It resides in an occluded conformation where the access to the binding site is sealed from both sides of the membrane. This structure harbors an ion binding region for $Al^{3+}$ located inside an aqueous pocket, which encompasses the predicted replacements of coordinating residues and which also contains an additional acidic amino acid that increases the negative charge density of the site to accommodate the larger net charge of the transported substrate.

## Results

### Functional characterization of SiNRAT

In the course of a previous phylogenetic analysis, we were able to spot seven homologues of the prototypic NRAT1 from *Oryza sativa* (XP_015625418.1, Uniprot: Q6ZG85) in Blast searches of eukaryotic sequence databases (*Ramanadane et al., 2022*). All identified proteins contain the described hallmarks of NRATs, which distinguishes this clade of the SLC11 family from the bulk of transition metal ion transporters (*Figure 1—figure supplement 1*). Differences include variations in the putative ion coordinating site, where the conserved DPGN motif on α1 of classical NRAMPs is changed to DPSN and the A/CxxM motif on α6 to AxxT (*Lu et al., 2018*; *Ramanadane et al., 2022*). Latter replaces the conserved methionine, whose sulfur atom is involved in transition metal ion coordination, with a threonine, which offers a hard oxygen ligand for metal ion interactions. To identify proteins that are suitable for structural and functional investigations, we have generated constructs of five NRAT homologues, each containing a C-terminal fusion of eGFP and an SBP tag used for affinity purification preceded by a 3C protease cleavage site. We have subsequently investigated their expression upon transient transfection of HEK293 cells followed by an initial biochemical characterization by fluorescence size exclusion chromatography (*Hattori et al., 2012*). These experiments have singled out the protein from *S. italica* (termed SiNRAT, XP_004952002.1) as construct with comparably high expression level and promising biochemical behavior. We have subsequently scaled up the expression of SiNRAT in suspension culture and purified it in the detergent *n*-dodecyl-β-D-maltopyranoside (DDM) for further studies (*Figure 1—figure supplement 2A, B*).

For a characterization of its transport properties, SiNRAT was reconstituted into proteoliposomes and subjected to previously established fluorescence-based assays that have permitted the functional investigation of other members of the SLC11 family (*Ehrnstorfer et al., 2014*; *Ehrnstorfer et al., 2017*; *Ramanadane et al., 2022*). We have initially studied the transport of $Mn^{2+}$, which is a common substrate of SLC11 transporters that can be monitored using the fluorophore calcein. Although, in a physiological context, SiNRAT presumably operates at mildly acidic extracellular conditions (i.e. at a pH <5.5), all transport experiments were carried out at neutral pH since this resulted in maximum proteoliposome stability and highest sensitivity of the employed fluorophores. The assay revealed a concentration-dependent quenching of calcein as a consequence of the influx of $Mn^{2+}$ into the proteoliposomes (*Figure 1A*). Transport is strongly dependent on the $Mn^{2+}$ concentration and saturates with an apparent $K_m$ of about 14 µM, reflecting the binding of the transported substrate to a saturable site (*Figure 1A and B*). We subsequently investigated impairment of $Mn^{2+}$ transport by the alkaline earth metal ions $Ca^{2+}$ and $Mg^{2+}$ and found a concentration-dependent competition by both ions, which is indicative for their specific binding to the protein (*Figure 1C and D*, *Figure 1—figure supplement 2C, D*). To investigate whether they would also be transported, we directly monitored their influx into proteoliposomes employing the fluorophores Fura-2 for the detection of $Ca^{2+}$ and Magnesium Green for $Mg^{2+}$. In both cases, we found a weak signal that saturates at µM concentrations, which reflects uptake of both ions (*Figure 1E and F*, *Figure 1—figure supplement 2E-H*).

After demonstrating that SiNRAT is capable of mediating the transport of divalent metal ions with poor selectivity, we were interested in the transport properties of $Al^{3+}$, which is the proposed physiological substrate. However, assaying this ion under in vitro conditions is complicated by the lack of suitable fluorophores and the fact that the chemical properties of $Al^{3+}$, as strong Lewis acid with an ionic radius of 0.53 Å, leads to the destabilization of liposomes even at low µM concentrations. We thus turned our initial attention to the trivalent cation $Ga^{3+}$, which is one period below $Al^{3+}$ in the same group of the periodic system, whose larger ionic radius (0.62 Å) and consequent weaker Lewis acidity is less disruptive for lipid bilayers. In an assay to investigate the interference with $Mn^{2+}$ transport, we found an inhibition by $Ga^{3+}$, which emphasizes its interaction with the transporter (*Figure 1G*). Conversely, we did not observe such competition with the prokaryotic transition metal ion transporter EcoDMT, excluding the action of $Ga^{3+}$ as non-specific inhibitor of the family (*Figure 1—figure supplement 2I*). Thus, despite the low solubility of $Ga^{3+}$ under the assayed conditions (*Yu and Liao, 2011*), which obscures a definitive assignment of its concentration dependence, our results suggest a specific effect of the ion on SiNRAT-mediated transport. Since equivalent studies with $Al^{3+}$ are prevented by its disruptive properties on membrane integrity, we used isothermal titration calorimetry (ITC) to investigate the binding of the ion to SiNRAT in detergent solution. These experiments showed a specific signal upon $Al^{3+}$ titration as a consequence of its interaction with the transporter (*Figure 1H*). However, at the neutral pH of the sample, the trivalent ion is at equilibrium with its hydroxyl complexes, which complicates data analysis (*Kiss, 2013*). The thermograms were thus best fitted to a sum of two binding constants, both in the low micromolar range (*Figure 1I*). Although precluding a definitive mechanistic interpretation, we use the ITC results of $Al^{3+}$ binding as a qualitative assay of its interaction with SiNRAT and show later that the event is directly associated with its binding to the site that is relevant for ion transport.

We then investigated whether metal ion transport would be accompanied by a concentration-dependent acidification of vesicles, which is expected in case of coupled proton symport as observed for the prokaryotic SLC11 transporter EcoDMT (*Figure 1J*; *Ehrnstorfer et al., 2017*). To this end, we have assayed the import of $H^+$ at increasing concentrations of the transported ions $Mn^{2+}$, $Ca^{2+}$, $Mg^{2+}$ and the putative substrate $Ga^{3+}$ but did in no case detect evidence for metal ion concentration-dependent decrease of ACMA fluorescence, despite the negative membrane potential established by an outwardly directed $K^+$ gradient, which would facilitate $H^+$ transport (*Figure 1K and L*, *Figure 1—figure supplement 2J, K*). These findings suggest that, akin to the $Mg^{2+}$ transporter EleNRMT, SiNRAT acts as uncoupled transporter that facilitates the bidirectional transport of metal ions with poor selectivity.

## Structural characterization of SiNRAT

The described transport properties of SiNRAT are unique among characterized members of the SLC11 family. To define the underlying molecular features, we set out to determine its three-dimensional

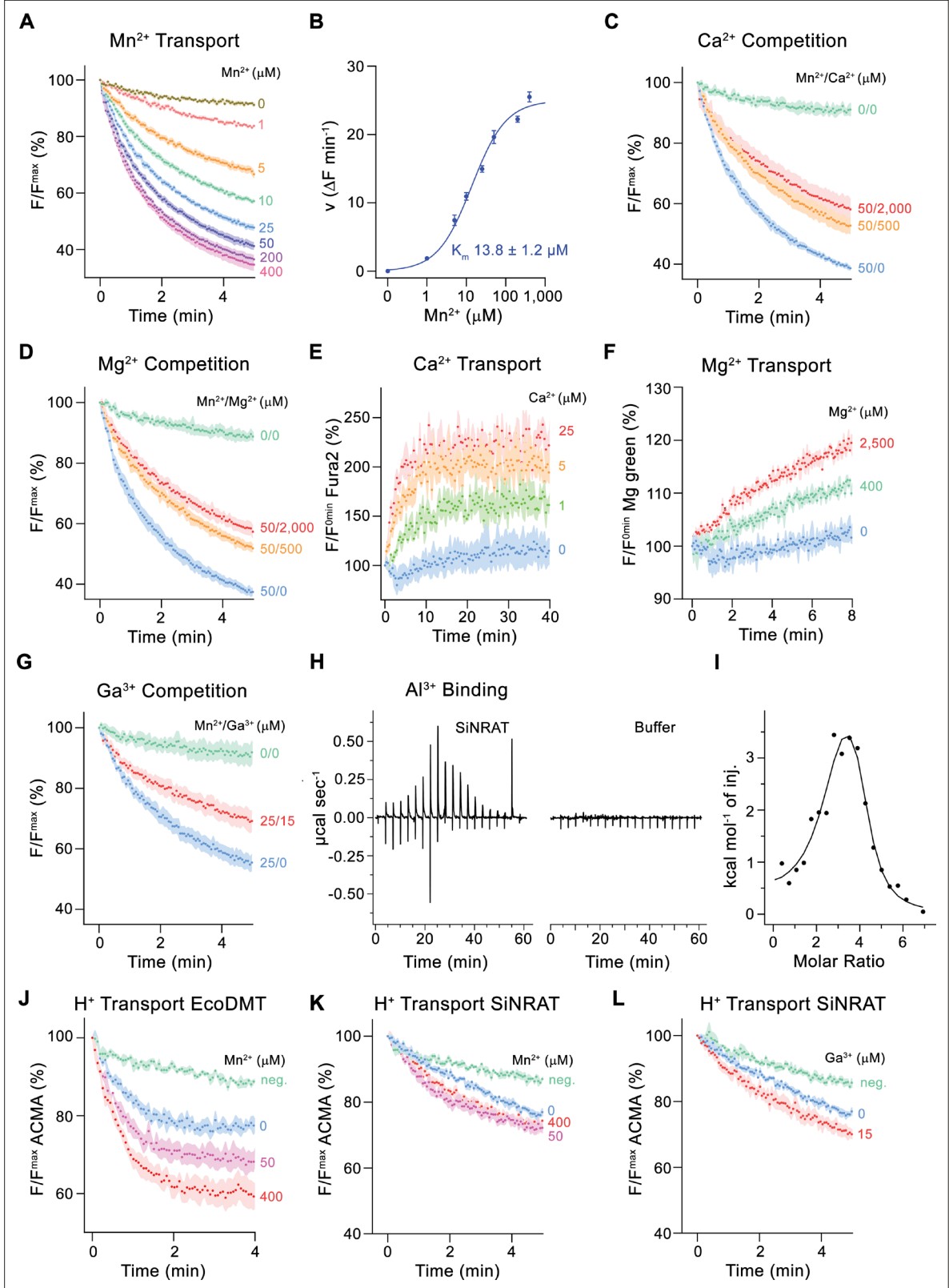

**Figure 1.** Functional characterization of SiNRAT. (**A**) SiNRAT-mediated $Mn^{2+}$ transport into proteoliposomes (six experiments from four independent reconstitutions). (**B**) $Mn^{2+}$ concentration dependence of transport. Initial velocities were derived from individual traces of experiments displayed in (**A**), the solid line shows the fit to a Michaelis–Menten equation with an apparent $K_m$ of $13.8 \pm 1.2$ µM (with the error based on a 95% confidence interval). (**C**) $Mn^{2+}$ transport in presence of $Ca^{2+}$ (three experiments from three independent reconstitutions for all conditions). (**D**) $Mn^{2+}$ transport in presence of

*Figure 1 continued on next page*

*Figure 1 continued*

$Mg^{2+}$ (three experiments from three independent reconstitutions for all conditions). (**E**) SiNRAT-mediated $Ca^{2+}$ import into proteoliposomes assayed with the fluorophore Fura2 trapped inside the liposome (three experiments from three independent reconstitutions for all conditions). (**F**) Assay of $Mg^{2+}$ import into proteoliposomes assayed with the fluorophore Magnesium Green (three experiments from three independent reconstitutions for all conditions). (**G**) $Mn^{2+}$ transport in presence of $Ga^{3+}$ (five experiments from two independent reconstitutions for conditions containing $Mn^{2+}$, four experiments for the condition 0 µM $Mn^{2+}$/0 µM $Ga^{3+}$). (**A, C, D, G**) Uptake of $Mn^{2+}$ was assayed by the quenching of the fluorophore calcein trapped inside the vesicles. (**H**) Thermograms of $Al^{3+}$ binding to SiNRAT (left) and buffer (right) obtained from isothermal titration calorimetry experiments. (**I**) Binding isotherm of $Al^{3+}$ was fitted to a sum of two binding constants with the binding isotherm depicted as solid line. (**J–L**) Assay of $H^+$ transport with the fluorophore ACMA. Experiments probing metal ion coupled $H^+$ transport into proteoliposomes upon addition of metal ions to the outside. (**J**) $H^+$ transport into proteoliposomes containing EcoDMT upon addition of $Mn^{2+}$ (three experiments from two independent reconstitutions for all conditions). $H^+$ transport into proteoliposomes containing SiNRAT upon addition of (**K**) $Mn^{2+}$ (four experiments from three independent reconstitutions) and (**L**) $Ga^{3+}$ (four experiments from three independent reconstitutions for SiNRAT with no substrate and empty liposomes with 15 µM $Ga^{3+}$ and three experiments from three independent reconstitutions for SiNRAT with 7.5 or 15 µM $Ga^{3+}$). (**A–G**) and (**J–L**) Panels show mean of indicated number of experiments, errors are s.e.m. (**A**), (**C**), (**D–G**), (**J–L**) Fluorescence is normalized to the value after addition of substrate (t=0). Applied ion concentrations are indicated. Negative controls (neg.) refer to empty liposomes in presence of 400 µM $Mn^{2+}$ or 15 µM $Ga^{3+}$, respectively.

The online version of this article includes the following source data and figure supplement(s) for figure 1:

**Source data 1.** Transport and ITC data of SiNRAT.

**Figure supplement 1.** Sequence alignment of the metal ion binding site of SLC11 family members.

**Figure supplement 2.** Purification and assay data.

**Figure supplement 2—source data 1.** Biochemistry and transport data of SiNRAT.

(3D) structure. Since the protein did not crystallize, we generated nanobody-based binders that, in complex with SiNRAT, increase the size of the protein sufficiently to permit its structure determination by cryo-electron microscopy (cryo-EM). These binders were obtained by immunization of alpacas with the purified protein and subsequently selected by phage display from a library generated from a blood sample (*Pardon et al., 2014*). The procedure allowed the identification of the nanobody Nb1$^{SiNRAT}$ (short Nb1) as promising interaction partner that does not dissociate during size exclusion chromatography, and which inhibits $Mn^{2+}$ transport in a concentration-dependent manner when added to the outside of liposomes (*Figure 2—figure supplement 1*). The incomplete inhibition of transport by Nb1 presumably reflects the mixed distribution of reconstituted SiNRAT residing in either inside-out or outside-out orientation, where the epitopes recognized by the nanobody are only accessible in one orientation. For the SiNRAT-Nb1 complex, we have prepared samples for cryo-EM. Since data recorded in the detergent DDM did not permit a 3D reconstruction at high resolution, we have reconstituted the complex into amphipols (*Kleinschmidt and Popot, 2014*) and collected a large dataset, which ultimately yielded a cryo-EM density map at 3.66 Å (*Figure 2—figure supplements 1–3*, *Table 1*, *Table 2*). Due to the challenges associated with the amphipol reconstitution process, the sample used for imaging did not contain transported ions. In case of trivalent ions this would have required a lowering of the pH to increase their solubility, which was incompatible with nanobody binding. The resulting map was of high quality and allowed the unambiguous interpretation with an atomic model revealing details of the 3D organization of SiNRAT (*Figure 2A*, *Figure 2—figure supplement 3*).

## SiNRAT structure and presumable ion interactions

The SiNRAT structure in complex with Nb1 is depicted in *Figure 2*. It shows a protein sharing the conserved architecture of the SLC11 family with a core consisting of a pair of structurally related repeats of five transmembrane spanning segments that are oriented in the membrane with opposite directions in an arrangement that was initially observed in the amino acid transporter LeuT (*Yamashita et al., 2005*). In this architecture, the unwound centers of the first helix of each repeat together constitute a substrate binding site (*Figure 2—figure supplement 4A*). The protein contains a total of 12 membrane-inserted helices akin to other eukaryotic family members and the prokaryotic EcoDMT (PDBID: 5M87) (*Ehrnstorfer et al., 2017*), whose Cα-positions superimpose with an RMSD of 2.66 Å (*Figures 2B and 3A*, *Figure 2—figure supplement 4B*). The nanobody binds to an extended interface located at the extracellular side. These interactions involve contacts of all three CDRs of the nanobody, which bridge the core domain of SiNRAT consisting of α-helices 1–10 with α-helices 11–12 located at its periphery, thereby contacting the loops α1–2, α5–6, α7–8 and residues on α11 and

**Table 1.** Cryo-electron microscopy (cryo-EM) data collection, refinement, and validation statistics.

| | Dataset 1<br>SiNRAT-NB1<br>(EMDB-17000)(PDB 8ONT) |
|---|---|
| Data collection and processing | |
| Microscope | FEI Titan Krios |
| Camera | Gatan K3 GIF |
| Magnification | 130,000 |
| Voltage (kV) | 300 |
| Electron exposure (e⁻/Å²) | 69.739/69.63 |
| Defocus range (μm) | –1.0 to –2.4 |
| Pixel size* (Å) | 0.651 (0.3255) |
| Initial number of micrographs (no.) | 16,530 |
| Initial particle images (no.) | 5,445,873 |
| Final particle images (no.) | 294,761 |
| Symmetry imposed | C1 |
| Map resolution (Å)<br>FSC threshold | 3.66<br>0.143 |
| Map resolution range (Å) | 2.8–7.0 |
| Refinement | |
| Model resolution (Å)<br>FSC threshold | 3.7<br>0.5 |
| Map sharpening b-factor (Å²) | –186.3 |
| Model vs Map CC (mask) | 0.78 |
| Model composition<br>Non-hydrogen atoms<br>Protein residue<br>Water<br>Ligand (PLC) | <br>4381<br>558<br>1<br>1 |
| B factors (Å2)<br>Protein<br>Water<br>Ligand | <br>205.39<br>129.77<br>126.19 |
| R.m.s. deviations<br>Bond lengths (Å)<br>Bond angles (°) | <br>0.003<br>0.386 |
| Validation<br>MolProbity score<br>Clashscore<br>Poor rotamers (%) | <br>2.39<br>31.28<br>0.22 |
| Ramachandran plot<br>Favored (%)<br>Allowed (%)<br>Disallowed (%) | <br>93.84<br>6.16<br>0 |

*Values in parentheses indicate the pixel size in super-resolution.

burying 1682 Å² of the combined molecular surface (*Figure 2—figure supplement 4C, D*). Unlike in EcoDMT, in this complex α11 and α12 are detached from the core domain on the extracellular part in a conformation that is presumably stabilized by Nb1 binding (*Figure 2B*, *Figure 2—figure supplement 4B, C*).

**Table 2.** Nanobody and primer sequences.

| Name | sequence |
| --- | --- |
| Nanobody 1 | CAGTGGCAGTTGGTGGAGTCTGGGGGAGGATTGGTGCA GGCTGGGGGCTCTCTGAGACTCTCCTGTGTGTAGGCTCTGG ACGCGCCTTCAGTAGCGGCGCCATGGGCTGGTTCCGCCA GACTCCAGGGCAGGAGCGTGAGTTTGTCGCAGCTATTAGC TGGAGTGGTGGTAGTACCGTCTATGCAGAGTCCGTGAAGG GCCGATTCACCATCTCCATGGACAACGCCAAGAACACGGT GTATCTGCGAATGAACAGCCTGCAACCTGAGGACACGGCC GTTTATTACTGTGCAGCCGGGACCAGTACATTCGCACTGCG TAGGTCCCCGGAATACTGGGGCAAAGGGACCCCGGTCACC GTCTCCAGT |
| D66A primer | for: CAT TGG CTT CCT GGC TCC TAG CAA CTT G rev: CAA GTT GCT AGG AGC CAG GAA GCC AAT G |
| S68A primer | for: GGC TTC CTG GAT CCT GCC AAC TTG GAA ACT G rev: CAG TTT CCA AGT TGG CAG GAT CCA GGA AGC C |
| N69A primer | for: CTT CCT GGA TCC TAG CGC CTT GGA AAC TGA CAT G rev: CAT GTC AGT TTC CAA GGC GCT AGG ATC CAG GAA G |
| T241A primer | for: CTA TTT GGC GCT ATC ATC GCA CCG TAC AAC TTG TTC rev: GAA CAA GTT GTA CGG TGC GAT GAT AGC GCC AAA TAG |
| T241V primer | for: CTA TTT GGC GCT ATC ATC GTA CCG TAC AAC TTG TTC TTG rev: CAA GAA CAA GTT GTA CGG TAC GAT GAT AGC GCC AAA TAG |

The observed structure resembles an inward-occluded state of the classical SLC11 transporter DraNRAMP (PDBID: 8E60, DraNRAMP$^{occ}$), where most of the structure retains a conformation observed in the inward-facing structure of the same protein (PDBID: 6D9W, DraNRAMP$^{inw}$) except for a movement of the intracellular part of α-helix 1 (α1a), which has rearranged to close the access path to the ion binding site (*Figure 3*, *Figure 3—figure supplement 1A, B*; *Bozzi et al., 2019b*; *Ray et al., 2023*). This resemblance is reflected in an RMSD of Cα atoms of 1.57 Å upon a superposition

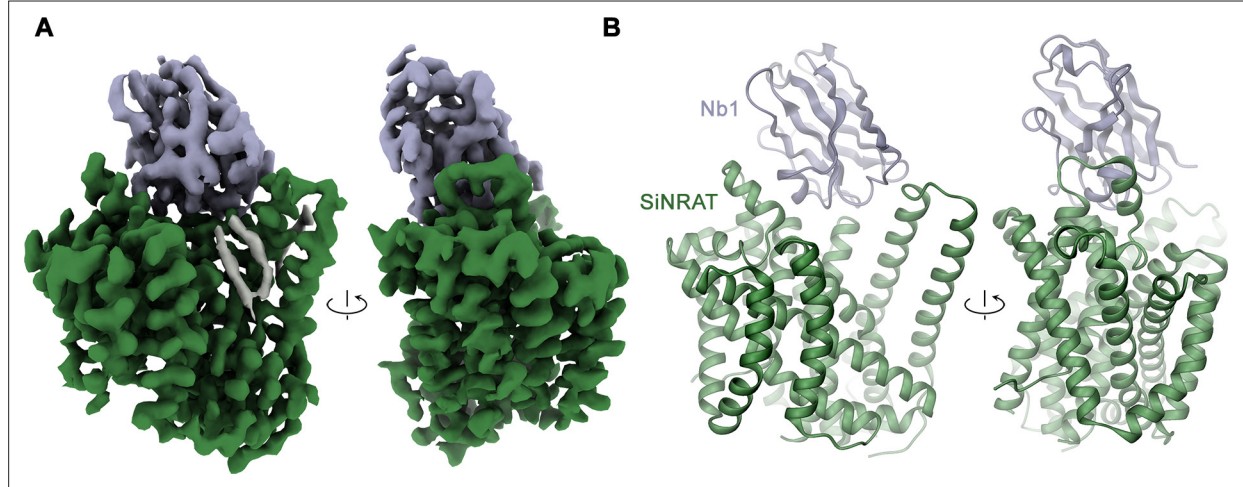

**Figure 2.** Structural characterization of the SiNRAT-Nb1 complex by cryo-electron microscopy (cryo-EM). (**A**) Cryo-EM density of SiNRAT in complex with Nb1 at 3.66 Å viewed from within the membrane at indicated orientations with the extracellular side on top and (**B**) ribbon representation of the complex in the same views. (**A**), (**B**) Proteins are shown in unique colors, the density of a bound lipid with uncertain orientation in (**A**) is shown in gray.

The online version of this article includes the following source data and figure supplement(s) for figure 2:

**Figure supplement 1.** Nanobody characterization.

**Figure supplement 1—source data 1.** Chromatograms, SDS PAGE gels and inhibition data.

**Figure supplement 2.** Cryo-electron microscopy (cryo-EM) reconstruction of the SiNRAT-Nb1 complex.

**Figure supplement 3.** Cryo-electron microscopy (cryo-EM) density of the SiNRAT-Nb1 complex.

**Figure supplement 4.** Structural features of the SiNRAT-Nb1 complex.

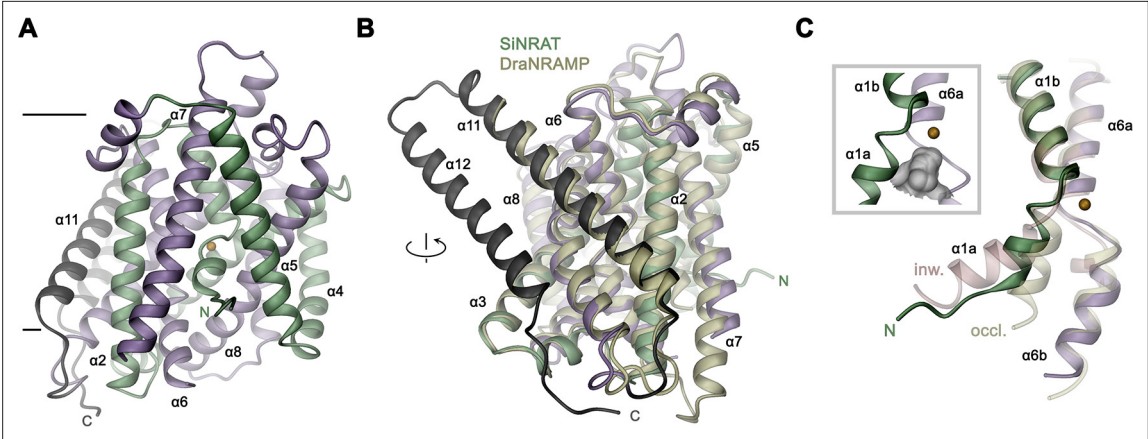

**Figure 3.** SiNRAT structure. (**A**) Ribbon representation of SiNRAT viewed from within the membrane with membrane boundaries indicated. (**B**) Superposition of SiNRAT with the protein DraNRAMP in an inward-occluded conformation. The orientation compared to A is indicated. (**C**) Ion binding site in relation to the α-helices 1 and 6. The same helices of the superimposed structures of DraNRAMP$^{occ}$ (PDBID: 8E30, occl.) and DraNRAMP$^{inw}$ (PDBID: 6D9W, inw.) are shown for comparison. Inset (left, framed) shows the region surrounding the binding site with the molecular surface of the partially closed intracellular cavity displayed in gray. (**A**), (**C**) The position of the ion binding site is indicated by an orange sphere. (**A–C**) The N- and C-terminal repeats (α1–5 and a6–10) are colored in green and indigo, respectively, the terminal helices α11 and α12 in dark gray.

The online version of this article includes the following figure supplement(s) for figure 3:

**Figure supplement 1.** Conformational properties of SLC11 transporters.

of SiNRAT with DraNRAMP$^{occ}$ and 1.74 Å with DraNRAMP$^{inw}$, whereas the RMSD of 2.71 Å compared to the outward-facing conformation of the same protein (PDBID: 6BU5, DraNRAMP$^{out}$) is considerably larger (*Figure 3C*, *Figure 3—figure supplement 1C*). As in DraNRAMP$^{occ}$, the shorter α1a of SiNRAT is located closer to α6b compared to DraNRAMP$^{inw}$, resulting in a narrowing of an aqueous cavity leading toward the supposed ion binding site, thereby separating this site from the intracellular environment (*Figure 3C*, *Figure 3—figure supplement 1D*). We thus assume that the SiNRAT structure represents an intermediate between both endpoints of a transporter, functioning by an alternate access mechanism, that is closer to an inward-facing state. In its conformation, the binding site is sealed to both sides of the membrane by a thin gate composed of a single layer of residues (i.e. Asp 66, Asp 140, Thr 241, Tyr 243, and Thr 346) toward the inside and a thick gate (encompassing several layers of residues and spanning about 15 Å) toward the outside (*Figure 3—figure supplement 1D*).

Although for the reasons described above, we have not explicitly added transported ions to our sample, we find residual cryo-EM density bridging the neighboring residues Asp 66 and Ser 68 which, as part of the supposed ion binding site, are both located on the unwound loop connecting α1a and α1b (*Figure 4A*). This density presumably either originates from water, the monovalent cation Na$^+$ that is contained in the sample at a concentration of 200 mM, or traces of divalent metal ions found in our solution. Due to its proximity to binding sites identified in structures of other family members, we use this position as reference to analyze plausible protein substrate interactions, which nevertheless should be considered as tentative. In this putative site, Asp 66 is particularly striking, since it is universally conserved and was shown to constitute a key component for ion coordination in the entire SLC11 family. In contrast, Ser 68 is unique to NRATs and substituted by a Gly in all other members (*Figure 1—figure supplement 1*). The comparison of the SiNRAT site with the ion interaction in DraNRAMP$^{occ}$ shows common features but also differences that may underlie the altered substrate preference of SiNRAT. In both structures, the water-filled paths to the metal ion binding site are sealed from both sides of the membrane and bound ions would thus be segregated from their aqueous environment, although the occlusion toward the intracellular solution is more pronounced in DraNRAMP$^{occ}$ (*Figures 3C, 4B and C*, *Figure 3—figure supplement 1B, D*). In its binding site, the interacting transition metal ion in DraNRAMP$^{occ}$ is predominantly enclosed by protein residues except for one contact with a small pocket harboring trapped water molecules (*Ray et al., 2023*). The side chains of Asp 56 and Asn 59 on α1, in conjunction with the backbone carbonyls of Ala 53 and Ala 227, tightly surround the bound Mn$^{2+}$. Additionally, the thioether group of Met 230 contributes a soft ligand for ion interactions (*Figure 4C*). Finally, a water-mediated interaction with the side chain of Gln

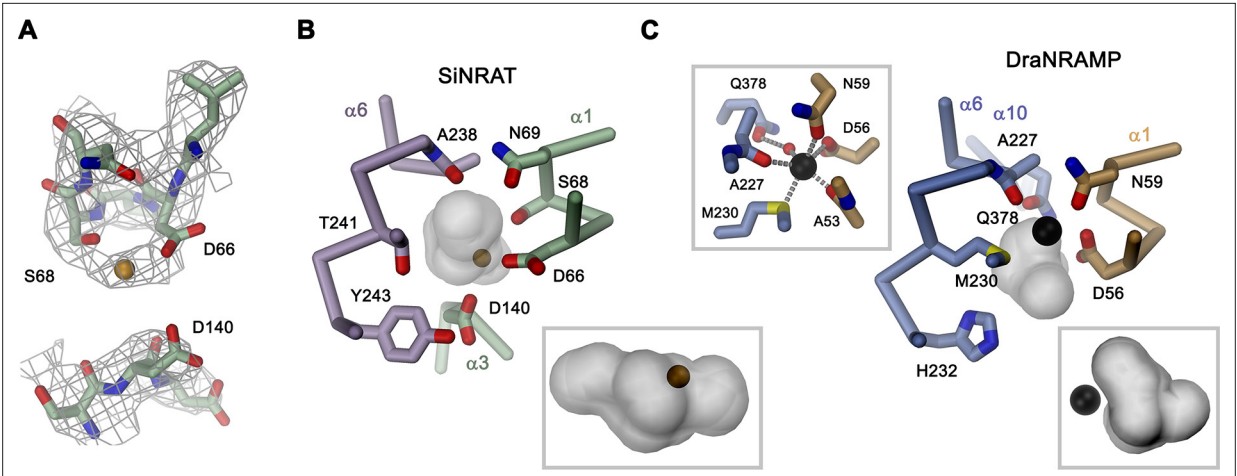

**Figure 4.** SiNRAT metal ion binding site. (**A**) Selected parts of the SiNRAT ion binding region with cryo-electron microscopy (cryo-EM) density superimposed. Top, loop connecting helices α1a and α1b. Residual density between Asp 66 and Ser 68 was used to define the presumable position of the ion binding site (with a bound ion represented as orange sphere). Bottom, density around Asp 140 located on α3. Comparison of the metal ion binding site in SiNRAT (**B**) and DraNRAMP^occ (PDBID: 8E60) (**C**) viewed from within the membrane. Shown are the presumed positions of bound metal ions (spheres) and the surrounding protein as Cα trace with selected interacting side and main chain atoms labeled and displayed as sticks. In both cases, the occluded and presumably water-filled cavities surrounding the bound ions, which bury a volume of 6071 Å³ in SiNRAT and 2404 Å³ in DraNRAMP are shown as gray surfaces. Insets (marked by a gray frame, bottom right) show the same cavities viewed from the cytoplasm. In (**C**), a second inset (top left) shows the coordination geometry of the Mn²⁺ ion bound to DraNRAMP. (**A–C**) Residues of the N- and C-terminal repeats of both proteins are shown in unique colors.

The online version of this article includes the following figure supplement(s) for figure 4:

**Figure supplement 1.** Residues of potential relevance for H⁺ transport.

378 on α10 completes the octahedral coordination of the ion (**Figure 4C**). In contrast, the equivalent region in SiNRAT contains an elongated, approximately cylindrical pocket of predominant polar character that is about 10 Å long and 4–5 Å wide and that encloses a three times larger volume than in DraNRAMP (**Figure 4B**). This pocket appears of sufficient size to mediate metal ion interactions with protein residues and surrounding water molecules. Despite the inability to pinpoint the exact location of bound ions, we find residues of the consensus binding site to line about half of this pocket (**Figure 4B**). These include the previously mentioned Asp 66 and Ser 68, both bordering residual density in our cryo-EM data and the conserved Asn 69, all located on α1. On α6, the interacting residues include the conserved Ala 238, whose backbone carbonyl points toward the pocket and Thr 241, which replaces the conserved methionine in NRAMP transporters and whose side chain could contribute a hard ligand for either direct or water-mediated ion interactions (**Figure 4B**). Distinct from DraNRAMP, the glutamine on α10 involved in ion binding is replaced by a serine (Ser 403), whose shorter side chain does not contact the aqueous pocket and thus would not be available for metal ion interactions. Instead, we find the acidic side chain of Asp 140 located on α3 to line the cavity, close to the predicted ion binding site, at a position that contains a threonine (Thr 130) in DraNRAMP. This acidic side chain, which is conserved in NRATs, would increase the negative charge density of the region by offering favorable coulombic interactions that stabilize the additional positive charge of Al³⁺ (**Figure 4B**, **Figure 1—figure supplement 1**). Thus, besides the replacement of the conserved methionine of the transition metal ion binding site, we find an expanded interaction region with increased polar character and negative charge density, which together may account for the altered substrate preference and stabilize trivalent cations.

Besides the altered ion selectivity, the second feature that distinguishes SiNRAT from classical NRAMPs is the apparent lack of H⁺ coupling, which is also shared by prokaryotic NRMTs of the same family (**Ramanadane et al., 2022**). Although the detailed mechanism of H⁺ coupling in the SLC11/NRAMP family is still a matter of debate, there are remarkable structural differences between classical H⁺-coupled metal ion transporters, SiNRAT, and the Mg²⁺ transporter EleNRMT (**Figure 4—figure supplement 1**). In NRAMPs, H⁺ transport was associated with acidic and basic residues located on the α-helices 1, 3, 6, and 9, most of which are replaced by hydrophobic side chains in NRMTs (**Bozzi**

*et al., 2019a*; *Bozzi et al., 2020*; *Ehrnstorfer et al., 2017*; *Ramanadane et al., 2022*; *Figure 1—figure supplement 1*, *Figure 4—figure supplement 1B, C*). In contrast, many of these residues are conserved in NRATs, which are evolutionarily closer to NRAMPs (*Figure 1—figure supplement 1*, *Figure 4—figure supplement 1A*). A pronounced difference between NRATs and NRAMPs concerns the substitution of a conserved histidine on α6b (i.e. His 232 in DraNRAMP) (*Ehrnstorfer et al., 2017*; *Lam-Yuk-Tseung et al., 2003*) with a tyrosine (Tyr 243 in SiNRAT), whereas the equivalent position in NRMTs is occupied by a tryptophane (i.e. W229 in EleNRMT) (*Figure 1—figure supplement 1*, *Figure 4—figure supplement 1*). However, the exact relation of the described structural features to the observed transport mechanism is still unclear and requires further investigation.

## Functional properties of SiNRAT binding site mutants

After defining the structure of the putative metal ion binding site of SiNRAT, we decided to study the effect of point mutants on the transport properties of the protein. To this end, we have put our focus on residues of the consensus binding site located on α-helices 1 and 6, whose equivalent positions were found to contribute to ion interaction in transition metal ion transporters of the SLC11 family (*Figure 5A*). All investigated mutants were purified and either reconstituted into liposomes for uptake studies of $Mn^{2+}$ or used for ITC experiments in their detergent-solubilized form. Upon mutation of the residues Asp 66 or Asn 69 to alanine, we found a severely compromised transport of $Mn^{2+}$, consistent with equivalent results obtained for other pro- and eukaryotic transition metal ion transporters (*Figure 5B and C*; *Bozzi et al., 2019b*; *Ehrnstorfer et al., 2014*; *Ehrnstorfer et al., 2017*; *Ramanadane et al., 2022*). We next turned our attention toward Thr 241, which has substituted the conserved methionine found in transition metal ion transporters to provide a potential hard ligand for direct or water-mediated metal ion interactions. In this case, we have mutated the residue to alanine to truncate the side chain and valine to preserve its volume but remove its polar character. In both cases, we found a similar strong negative impact of the respective mutations on $Mn^{2+}$ transport (*Figure 5D and E*). For this residue, we were also interested in whether a mutation would affect interactions with trivalent cations and thus investigated binding of $Al^{3+}$ to the mutant T241V by ITC. Unlike WT, we did not detect any specific heat exchange, which emphasizes the importance of the residue for $Al^{3+}$ interactions (*Figure 5F*). The absence of a detectable signal in T241V also underlines the correspondence of the ITC data of WT to $Al^{3+}$ binding to the consensus site (*Figure 1H*). Finally, we have studied a mutation of Ser 68 to alanine, which concerns a residue that is unique to NRATs and which was found in the SiNRAT structure to contribute to potential ion interactions (*Figure 4A*, *Figure 1—figure supplement 1*). Unlike previous mutations, we did in this case not observe an obvious effect on $Mn^{2+}$ transport, which proceeds with similar kinetics as WT (*Figure 5G and H*) and whose transport is competed by the addition of $Mg^{2+}$ and $Ca^{2+}$ in a similar manner (*Figure 5I and J*). Unlike WT, we did not detect a competition by $Ga^{3+}$ in equivalent experiments (*Figure 5K*). We thus set out to investigate the binding of $Al^{3+}$ by ITC but did not find any indication for its interaction with the mutant (*Figure 5L*). Taken together, our experiments on the SiNRAT mutant S68A showed distinct phenotypes for the interaction with di- and trivalent cations, which points toward structural differences in their coordination with more stringent requirements for $Ga^{3+}$ and $Al^{3+}$ binding.

## Discussion

Plants growing on acidic soils have to cope with elevated concentrations of $Al^{3+}$ which, as strong Lewis acid, forms tight interactions with phospholipids and carbohydrates, thereby compromising the integrity of the membrane and the cell wall (*Chauhan et al., 2021*). As one of the mechanisms to combat $Al^{3+}$ toxicity, certain plants have evolved a branch of the NRAMP family with altered substrate preference termed NRATs, which are capable of transporting the trivalent cation (*Xia et al., 2010*). In our study, we have determined the structure and transport properties of the NRAT of *Setaria italica* termed SiNRAT. Using fluorescence-based in vitro transport assays from proteoliposomes containing the purified transporter, our study has characterized SiNRAT as a protein with broad substrate specificity. We have shown transport of different divalent cations such as $Mn^{2+}$, $Ca^{2+}$, and $Mg^{2+}$ with micromolar $K_m$ (*Figure 1A–F*, *Figure 1—figure supplement 2F, H*). Although the direct assay of transport of trivalent ions such as $Ga^{3+}$ and $Al^{3+}$ was prohibited for technical reasons, we have demonstrated the interaction of SiNRAT with both ions, which defines them as plausible substrates (*Figure 1G–I*).

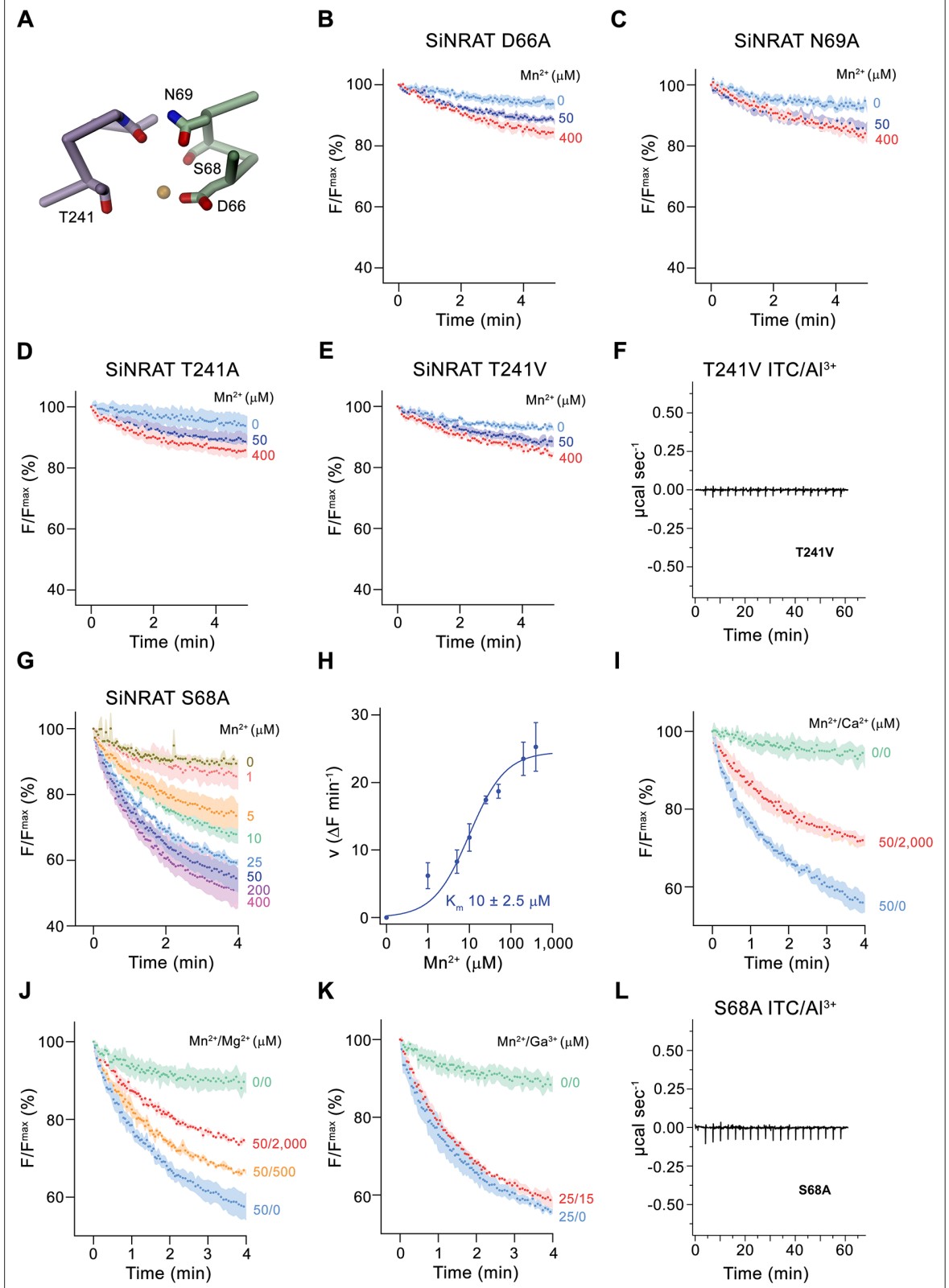

**Figure 5.** Functional properties of binding site mutants. (**A**) Cα trace of the metal ion binding region of SiNRAT with selected residues involved in ion coordination shown as sticks. (**B–E**) $Mn^{2+}$ transport into proteoliposomes of SiNRAT metal ion binding site mutants. (**B**) D66A (four experiments from three independent reconstitutions for all conditions); (**C**) N69A (three experiments from three independent reconstitutions); (**D**) T241A (four experiments from three independent reconstitutions for all conditions); (**E**) T241V (three experiments from three independent reconstitutions for

*Figure 5 continued on next page*

*Figure 5 continued*

all conditions). (**F**) Thermogram of $Al^{3+}$ titrated to the SiNRAT mutant T241V obtained from isothermal titration calorimetry experiments. (**G**) $Mn^{2+}$ transport into proteoliposomes containing the SiNRAT mutants S68A (three experiments from two independent reconstitutions for all conditions). (**H**) $Mn^{2+}$ concentration dependence of transport by the mutant S68A. Data show mean of initial velocities derived from individual traces of experiments displayed in (**G**), errors are s.e.m., the solid line shows the fit to a Michaelis–Menten equation with an apparent $K_m$ of 10±2.5 µM. (**I–K**) $Mn^{2+}$ transport in presence of other multivalent cations show interaction of the mutated binding site with $Ca^{2+}$ and $Mg^{2+}$ but not with $Ga^{3+}$ at indicated ion concentrations. (**I**) $Ca^{2+}$ (three experiments from two independent reconstitutions for all conditions); (**J**) $Mg^{2+}$ (three experiments from two independent reconstitutions); (**K**) $Ga^{3+}$ (three experiments from two independent reconstitutions). (**L**) Thermogram of $Al^{3+}$ titrated to the SiNRAT mutant S68A obtained from isothermal titration calorimetry experiments. (**B–E**), (**G**), (**I–K**) Uptake of $Mn^{2+}$ was assayed by the quenching of the fluorophore calcein trapped inside the vesicles. Panels show mean of indicated number of experiments, errors are s.e.m. Fluorescence is normalized to the value after addition of substrate (t=0). Applied ion concentrations are indicated.

The online version of this article includes the following source data for figure 5:

**Source data 1.** Transport and ITC data of SiNRAT mutants.

However, due to their poor solubility under the assay conditions, these results are tentative and should be taken with caution. The broad substrate selectivity is consistent with previous experiments on the metal ion transporters EcoDMT and DraNRAMP (*Bozzi et al., 2016*; *Ehrnstorfer et al., 2017*) and the $Mg^{2+}$ transporter EleNRMT (*Ramanadane et al., 2022*). Attempts to change the transport properties of these proteins by mutagenesis, while deteriorating selectivity, have usually retained the capability of mutants to permeate $Mn^{2+}$ (*Bozzi et al., 2016*; *Ramanadane et al., 2022*), whose transport appears to have lower structural requirements compared to other divalent cations. By the placement of a methionine contributing a soft ligand for transition metal ion interaction, the substrate binding site of classical NRAMPs has evolved to prevent interactions with alkaline earth metal ions (*Bozzi et al., 2016*; *Figure 6A*). This is illustrated in the mutation of this residue to alanine, which has turned $Ca^{2+}$ into a substrate without strongly affecting $Mn^{2+}$ transport (*Bozzi et al., 2016*; *Ramanadane et al., 2022*). Similarly, NRMTs transport the small alkaline earth metal ion $Mg^{2+}$ by increasing the volume of their binding site to interact with the hydrated cation, without compromising $Mn^{2+}$ transport (*Ramanadane et al., 2022*; *Figure 6B*). A comparable feature is observed for SiNRAT, where the protein

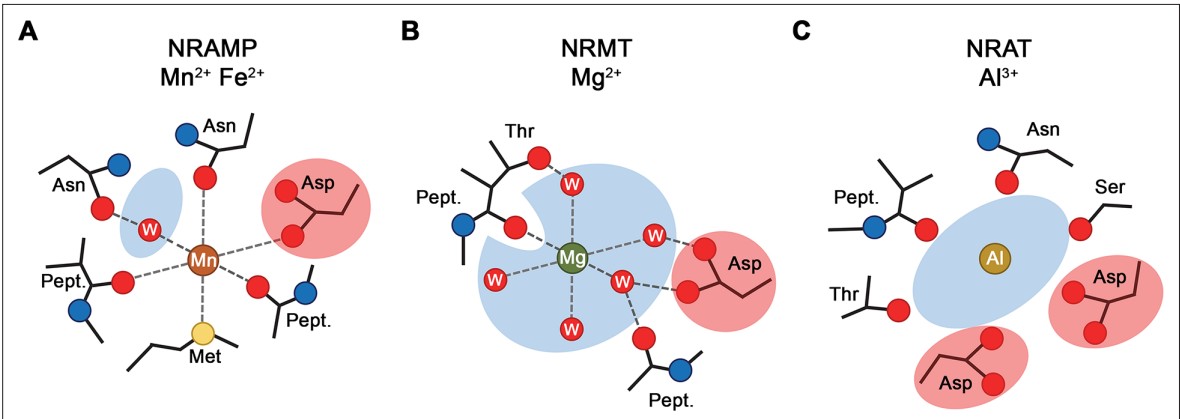

**Figure 6.** Selectivity and metal ion coordination in different clades of the SLC11 family. (**A**) Coordination of $Mn^{2+}$ in transition metal ion transporters (NRAMPs) of the SLC11 family as defined in the occluded conformation of the prokaryotic DraNRAMP (PDBID: 8E60). The octahedral coordination geometry is well defined. The largely dehydrated metal ion is predominantly forming direct protein interactions except for one interaction that is mediated by a bound water molecule. The thioether of a methionine serves as soft ligand in transition metal ion coordination. (**B**) Putative $Mg^{2+}$ interaction in an NRAMP-related $Mg^{2+}$ transporter (NRMT). The interactions were obtained from the inward-facing conformation of the protein EleNRMT (PDBID: 7QIA). The position of the metal ion was defined by bound $Mn^{2+}$ obtained from anomalous X-ray scattering experiments, coordinating water molecules are modeled. Owing to the larger volume of the binding site, the ion has retained most of its first coordination shell and the majority of ion-protein interactions are presumably indirect. (**C**) Putative coordination of $Al^{3+}$ in the transporter SiNRAT. The binding position was inferred from residual density in the SiNRAT cryo-electron microscopy (cryo-EM) map. The ion is placed in an aqueous cavity and likely undergoes interactions with the protein that are either direct or mediated by water molecules. A second presumably deprotonated aspartate increases the negative charge density in the binding site and thus stabilizes the greater charge density of the trivalent metal ion. (**A–C**) Interacting waters (**W**) are labeled, an aqueous cavity is indicated in blue, negatively charged groups involved in metal ion coordination are highlighted in red. Pept. refers to the interaction with the carbonyl of a peptide bond.

appears to have evolved to transport trivalent cations without imposing a strong selection against divalent metal ions (*Figure 1*). The SiNRAT structure has shed light into the basis for the altered selectivity of a transporter, whose evolutionary distance to classical NRAMPs is much smaller than to NRMTs (*Lu et al., 2018*; *Ramanadane et al., 2022*). Although, for technical reasons, our structure did not contain trivalent ions, we assigned their presumable binding region based on the structural equivalence to the transition metal ion transporters DraNRAMP and ScaDMT and the $Mg^{2+}$ transporter EleNRMT (*Ehrnstorfer et al., 2014*; *Ramanadane et al., 2022*; *Ray et al., 2023*). Residual density at the supposed location in the SiNRAT map provides further evidence for this assignment. Similar to NRMTs, and distinct from classical NRAMPs, the ion binding site of SiNRAT is embedded in a larger aqueous cavity, which offers direct metal ion interactions with protein residues and others that are mediated via hydrating waters (*Figure 4B and C*, *Figure 6C*; *Ramanadane et al., 2022*). Unlike the inward-facing structure of EleNRMT, in the structure of SiNRAT, this cavity is sealed from both sides of the membrane (*Figures 2C and 3B*). Within the canonical binding site of SiNRAT located on α1 and α6, several residues have been replaced to provide additional interactions and the presence of an extra acidic residue on α3 in interaction distance has presumably increased the negative charge density to compensate for the higher net charge of the transported ion (*Figure 4B*). In light of the described features, a proposed conversion of a classical plant NRAMP facilitating the transport of transition metals by the mutation of the consensus binding motif on α1 and α6 is likely insufficient to convert the protein into an efficient $Al^{3+}$ transporter (*Lu et al., 2018*). A related attempt to convert $Mg^{2+}$ into a transported substrate of the prokaryotic metal ion transporter EcoDMT by introducing corresponding mutations found in NRMTs was unsuccessful (*Ramanadane et al., 2022*), which emphasizes the relevance of residues outside a narrow signature region to confer substrate selectivity.

The observed broad substrate specificity of SiNRAT contrasts a previous study on the prototypic transporter NRAT1 from rice which, upon overexpression in yeast, was proposed to selectively transport $Al^{3+}$ but not the transition metal ion $Mn^{2+}$ (*Xia et al., 2010*). This apparently contradicting observation could reflect the distinct ways of how transport was assayed in both studies. We thus remain cautious with the assignment of substrates transported by NRATs in a physiological context, as the protein might not be efficient in concentrating $Mn^{2+}$ inside cells as a consequence of the absent coupling to an energy source. Still, in light of our data it is unclear how $Al^{3+}$ transport would proceed in presence of other divalent ions. Despite these ambiguities, our findings are in general agreement with a previous proposal that the NRAT-mediated uptake of $Al^{3+}$ into cells and its subsequent detoxification by complexation with organic anions and import into the vacuolar compartment would decrease its ability to interfere with cell wall integrity (*Xia et al., 2010*). In summary, our study has provided the first structural insight into the transport of trivalent cations and it revealed the architecture of a protein that is part of the defense against $Al^{3+}$ toxicity in plants, which continues to be a large challenge for farming on acidic soil. Since it was previously shown that the increased expression of proteins involved in the regulation of aluminum toxicity can lead to better aluminum tolerance (*Maron et al., 2013*), it will be interesting to investigate whether the overexpression of NRATs would increase the tolerance of plants to elevated $Al^{3+}$ levels.

# Materials and methods
## Expression and purification of EcoDMT

EcoDMT and its mutants were expressed and purified as described previously (*Ehrnstorfer et al., 2017*). Briefly, the vector EcoDMT-pBXC3GH was transformed into *E. coli* MC1061 cells and a preculture was grown in TB-Gly medium overnight using ampicillin (100 µg/ml) as a selection marker for all expression cultures. The preculture was used at a 1:100 volume ratio for inoculation of TB-Gly medium. Cells were grown at 37°C to an $OD_{600}$ of 0.7–0.9, after which the temperature was reduced to 25°C. Protein expression was induced by addition of arabinose to a final concentration of 0.004% (wt/vol) for 12–14 hr at 18°C. Cells were harvested by centrifugation for 20 min at 5000 × *g*, resuspended in buffer RES (200 mM NaCl, 50 mM KPi pH 7.5, 2 mM $MgCl_2$, 40 µg/ml DNAseI, 10 µg/ml lysozyme, 1 µM leupeptin, 1 µM pepstatin, 0.3 µM aprotinin, 1 mM benzamidine, 1 mM PMSF) and lysed using a high-pressure cell lyser (Maximator HPL6). After low-speed centrifugation (10,000 × *g* for 20 min), the supernatant was subjected to a second centrifugation step (200,000 × *g* for 45 min). The pelleted membrane was resuspended in buffer EXT (200 mM NaCl, 50 mM KPi pH 7.5, 10% glycerol)

at a concentration of 0.5 g of vesicles per ml of buffer. The purification of EcoDMT and its mutants was carried out at 4°C. Isolated membrane fractions were diluted 1:2 in buffer EXT supplemented with protease inhibitors (Roche cOmplete EDTA-free) and 1% (wt/vol) $n$-decyl-β-D-maltopyranoside (DM, Anatrace). The extraction was carried out under gentle agitation for 2 hr at 4°C. The lysate was cleared by centrifugation at 200,000 × $g$ for 30 min. The supernatant supplemented with 15 mM imidazole at pH 7.5 was subsequently loaded onto NiNTA resin and incubated for at least 1 hr under gentle agitation. The resin was washed with 20 column volumes (CV) of buffer W (200 mM NaCl, 20 mM HEPES pH 7, 10% glycerol, 50 mM imidazole pH 7.5, 0.25% DM) and the protein was eluted with buffer ELU (200 mM NaCl, 20 mM HEPES pH 7, 10% glycerol, 200 mM imidazole pH 7.5, 0.25% DM). Tag cleavage proceeded by addition of HRV 3C protease at a 3:1 molar ratio and dialysis against buffer DIA (200 mM NaCl, 20 mM HEPES pH 7, 10% glycerol, 0.25% DM) for at least 2 hr at 4°C. After application to NiNTA resin to remove the tag, the sample was concentrated by centrifugation using a 50 kDa molecular weight cut-off concentrator (Amicon) and further purified by size exclusion chromatography using a Superdex S200 column (GE Healthcare) pre-equilibrated with buffer SEC (200 mM NaCl, 20 mM HEPES pH 7, 0.25% DM). The peak fractions were pooled, concentrated, and used directly for further experiments.

## Cell culture

HEK293S GnTI⁻ and HEK293T cells were obtained from ATCC. HEK293S GnTI⁻ cells were grown in HyCell HyClone Trans Fx-H media supplemented with 1% FBS, 4 mM L-glutamine, 100 U/ml penicillin and 100 µg/ml streptomycin, and 1.5 g/l Kolliphor-188 at 37°C and 5% $CO_2$. Adherent HEK293T cells were grown in DMEM-Glutamax, supplemented with 10% FBS, 100 U/ml penicillin and 100 µg/ml streptomycin. All cell lines were regularly tested for prevention of mycoplasma contamination.

## Expression screening of NRAT homologues

The DNA coding for NRATs from different plants (i.e. the proteins from *O. sativa*, Uni prot: Q6ZG85; *D. oligosanthes*, NCBI: OEL35611.1; *S. bicolor*, NCBI: XP_002451480.2; *S. italica*, NCBI: XP_004952002.1; *Z. mays*, NCBI: PWZ19830.1) was synthesized by Genscript and cloned into a pcDNA3.1 vector (Invitrogen) using the FX cloning technique (*Geertsma and Dutzler, 2011*) yielding a final construct where each NRAT1 contains an HRV 3C protease cleavage site, eGFP, a Myc tag, and a streptavidin binding protein (SBP) attached to either its N- or C-terminus. Adherent HEK 293T cells (ATCC CRL-1573) were used for expression screening of plant NRAT1 homologues. Cells were grown on plates in DMEM-Glutamax High Glucose supplemented with 10% FBS, 100 U/ml penicillin, and 100 µg/ml streptomycin. Cells were grown and split to 60–70% confluency 1 day prior to transfection. Cells were transiently transfected using polyethyleneimine (PEI) 25 K. Briefly, for each 10 ml plate, 10 µg of DNA in 500 µl of non-supplemented DMEM was mixed with 40 µg of PEI 25 K in 500 µl of non-supplemented DMEM. After 15 min incubation, the transfection mixture was added to the plate together with 3 mM valproic acid. Cells were grown at 37°C and 5% $CO_2$ for 48–60 hr and harvested by centrifugation at 1000 × $g$ for 15 min. Pellets were resuspended in buffer LYS (200 mM NaCl, 20 mM HEPES pH 7, 10% glycerol, 2 mM $MgCl_2$, 40 µg/ml DNAse I, protease inhibitors [Roche Complete, EDTA-free], and 1% DDM). The lysate was incubated at 4°C for 1.5 hr under gentle agitation and clarified by centrifugation at 200,000 × $g$ for 15 min. The clarified whole-cell extract was subjected to fluorescent size exclusion chromatography (Agilent HPLC) using a Superdex S200 column (GE Healthcare) and detected with a fluorescence detector set to $\lambda_{ex}$ = 489 nm/ $\lambda_{em}$ = 510 nm. SiNRAT from *S. italica* (Uniprot KB K3YRE7) with a C-terminal tag was selected as the biochemically best behaving homologue based on the monodispersity of the eluted peak (*Hattori et al., 2012*).

## Expression and purification of SiNRAT

A plasmid carrying the sequence coding for the NRAT from the plant *S. italica* (SiNRAT) (Uniprot KB K3YRE7) cloned into a pcDNA3.1 vector (Invitrogen) containing an HRV 3C protease cleavage site, eGFP, a Myc and an SBP tag attached to its C-terminus was used for all further experiments. All mutants were generated using the QuickChange site-directed mutagenesis method (Agilent) (*Table 2*). SiNRAT was expressed by transient transfection using HEK293S GnTI⁻ cells diluted to 0.6–0.8·10⁶ cells/ml 1 day prior to transfection. DNA for transfection was purified from *E. coli* MC1061 cell line using the Nucleobond Xtra Maxi Kit (Macherey-Nagel); 1.3 µg DNA was used per 10⁶ cells. For that purpose,

the DNA was diluted to 16 µg/ml in a volume of non-supplemented DMEM corresponding to 1/14 of the transfected cell culture volume. In parallel, PEI-max 40 kDA was diluted to 0.04 mg/ml in the same volume of non-supplemented DMEM. The two solutions were mixed and after 15–20 min incubation, the final transfection mixture was added to the suspension cells. Finally, the cell culture was supplemented with 3 mM valproic acid. After 48–60 hr incubation, cells were harvested by centrifugation at $1500 \times g$ for 20 min and washed with PBS. All following steps of the purification were carried out at 4°C. The cell pellets were resuspended in buffer RES (200 mM NaCl, 50 mM KPi pH 7.5, 10% glycerol, 2 mM $MgCl_2$, 40 µg/ml DNAse I, and protease inhibitor [Roche Complete, EDTA-free]). The protein was extracted by adding DDM (Anatrace) to a final concentration of 2%. After an incubation for 2 hr under gentle agitation, the lysate was clarified by centrifugation at $200,000 \times g$ for 30 min. The supernatant was loaded on Streptactin Superflow affinity resin (IBA Lifesciences) and incubated for 2 hr under gentle agitation. The resin was washed with 25 CV of buffer W (200 mM NaCl, 20 mM HEPES pH 7, 10% glycerol, 0.04% DDM) and eluted with 20 ml of buffer ELU (buffer W supplemented with 5 mM D-desthiobiotin). The eGFP-Myc-SBP tag was removed by adding HRV 3C protease at a 1:3 molar ratio with SiNRAT and incubated on ice for 1 hr. The protein was concentrated by centrifugation to 500 µl using a 50 kDa molecular weight cut-off concentrator (Amicon) and further purified by size exclusion chromatography on a Superdex S200 column (GE Healthcare) pre-equilibrated with buffer SEC (200 mM NaCl, 20 mM HEPES pH 7, 0.04% DDM).

## Isothermal titration calorimetry

A MicroCal ITC200 system (GE Healthcare) was used for ITC experiments. All titrations were performed at 6°C in buffer SEC (200 mM NaCl, HEPES pH 7, 0.04% DDM). The syringe was filled with 1 mM $AlCl_3$ and the titration was initiated by the sequential injection of 2 µl aliquots into the cell filled with SiNRAT or mutants at a concentration of 30 µM. Data were analyzed with the Origin ITC analysis package and WT protein data were fitted using the integrated two sides model. Each experiment was repeated with independently purified protein samples with similar results. Experiments with buffer not containing any protein were performed as controls.

## Reconstitution of SiNRAT and EcoDMT into proteoliposomes

SiNRAT and its mutants were reconstituted into detergent destabilized liposomes (*Geertsma et al., 2008*). The lipid mixture consisting of POPE, POPG, and EggPC (Avanti Polar Lipids) at a weight ratio of 3:1:1 was washed with diethylether and dried under a nitrogen stream and by exsiccation overnight. The dried lipids were resuspended into 100 mM KCl and 20 mM HEPES pH 7. After three freeze-thaw cycles, the lipids were extruded through a 400 nm polycarbonate filter (Avestin, LiposoFast-BAsic) and aliquoted into samples at a concentration of 45 mg/ml. The extruded lipids were destabilized by addition of Triton X-100 and the protein was reconstituted at a protein to lipid ratio of 1:50. After several incubation steps with biobeads SM-2 (Bio-Rad), the proteoliposomes were harvested the next day, resuspended in 100 mM KCl and 20 mM HEPES pH 7, and stored at –80°C.

## Fluorescence-based substrate transport assays

The proteoliposomes were incubated with buffer IN (100 mM KCl, 20 mM HEPES pH 7, containing either 250 µM Calcein (Invitrogen), 100 µM Fura-2 (Thermo Fisher Scientific), or 400 µM MagGreen (Thermo Fisher Scientific) depending on the assayed ion). After three cycles of freeze-thawing, the liposomes were extruded through a 400 nm filter and the proteoliposomes were centrifuged and washed three times with buffer IN not containing any fluorophores. The proteoliposomes were subsequently diluted to 0.025 mg/ml in buffer OUT (100 mM NaCl, 20 mM HEPES pH 7) and aliquoted in 100 µl batches into a 96-well plate (Thermo Fisher Scientific) and the fluorescence was measured every 4 s using a Tecan Infinite M1000 fluorimeter. An inside negative membrane potential was generated by addition of valinomycin to a final concentration of 100 nM. After 20 cycles of incubation, the substrate was added at different concentrations. The transport reaction was terminated by addition of calcimycin or ionomycin at a final concentration of 100 nM. The initial rate of transport was calculated by linear regression within the initial 100 s of transport after addition of the substrate. The calculated values were fitted to a Michaelis-Menten equation. Fura-2 was used to assay the transport of $Ca^{2+}$. For detection, the ratio between calcium bound Fura-2 ($\lambda_{ex}$ = 340 nm; $\lambda_{em}$ = 510 nm) and unbound Fura-2 ($\lambda_{ex}$ = 380 nm; $\lambda_{em}$ = 510 nm) was monitored during kinetic measurements. Calcein was used

to assay the transport of $Mn^{2+}$ ($\lambda_{ex}$ = 492 nm; $\lambda_{em}$ = 518 nm) and Magnesium Green was to assay the transport of $Mg^{2+}$ ($\lambda_{ex}$ = 506 nm; $\lambda_{em}$ = 531 nm).

For assaying coupled proton transport, the proteoliposomes were mixed with buffer 1 (100 mM KCl, 5 mM HEPES pH 7, 50 μM ACMA). After clarification by sonication, the proteoliposomes were diluted to a concentration of 0.15 mg/ml in buffer 2 (100 mM NaCl, 5 mM HEPES pH 7) and aliquoted into batches of 100 μl into a flat black 96-well plate (Thermo Fisher Scientific). The fluorescence was measured every 4 s using a Tecan Infinite M1000 fluorimeter ($\lambda_{ex}$ = 412 nm; $\lambda_{em}$ = 482 nm). The transport reaction was initiated after addition of the substrate and valinomycin at a final concentration of 100 nM. The reaction was terminated by addition of CCCP at 100 nM. For experiments involving $Ga^{3+}$, $GaI_3$ was used. For compensation, NaI was added to reach a final I-concentration of 45 μM, which did not exert any effect on the measurement.

## Generation of nanobodies specific to SiNRAT

SiNRAT-specific nanobodies were generated using a method described previously (*Pardon et al., 2014*). Briefly, an alpaca was immunized four times with 200 μg of detergent-solubilized SiNRAT (in intervals of four weeks). Two days after the final injection, the peripheral blood lymphocytes were isolated, and the RNA total fraction was extracted and converted into cDNA by reverse transcription. The nanobody library was amplified and cloned into the phage display pDX vector (Invitrogen). After two rounds of phage display, ELISA assays were performed on the periplasmic extract of 198 individual clones. A single nanobody was identified. For all steps of phage display and ELISA, SiNRAT was chemically biotinylated using an EZ-Link NHS-PEG4 Biotinylation Kit (Thermo Fisher Scientific) and bound to neutravidin-coated plates (*Fairhead and Howarth, 2015*) . The biotinylation was confirmed by total mass spectrometry analysis.

## Expression and purification of nanobodies and preparation of the SiNRAT-Nb1 complex

Nb1 was cloned into the pBXNPHM3 vector (Invitrogen) using the FX cloning technique. The expression construct contained the nanobody with a fusion of a pelB sequence, a 10-His tag, a maltose binding protein (MBP) and an HRV 3C protease site to its N-terminus. The vector was transformed into *E. coli* MC1061 cells and a preculture was grown in TB-Gly medium overnight using ampicillin (100 μg/ml) as a selection marker in all expression cultures. The preculture was used for inoculation of TB-Gly medium at a 1:100 volume ratio. Cells were grown to an $OD_{600}$ of 0.7–0.9 and the expression was induced by the addition of arabinose to a final concentration of 0.02% (wt/vol) for 4 hr at 37°C. Cells were harvested by centrifugation for 20 min at 5000 × $g$, resuspended in buffer A (200 mM KCl, 50 mM KPi pH 7.5, 10% glycerol, 15 mM imidazole pH 7.5, 2 mM $MgCl_2$, 1 μM leupeptin, 1 μM pepstatin, 0.3 μM aprotinin, 1 mM benzamidine, 1 mM PMSF) and lysed using a high-pressure cell lyser (Maximator HPL6). The purification was carried out at 4°C. The lysate was cleared by centrifugation at 200,000 × $g$ for 30 min. The supernatant was loaded onto NiNTA resin and incubated for at least 1 hr under gentle agitation. The resin was washed with 20 CV of buffer B (200 mM KCl, 20 mM Tris-HCl pH 8, 10% glycerol, 50 mM imidazole, pH 7.5). The protein was eluted with buffer C (200 mM KCl, 20 mM Tris-HCl pH 8, 10% glycerol, 300 mM imidazole). The tag was cleaved by addition of HRV 3C protease at a 3:1 molar ratio and dialyzed against buffer D (200 mM KCl, 20 mM Tris-HCl pH 8, 10% glycerol) for at least 2 hr at 4°C. After application of NiNTA resin to remove the tag, the sample was concentrated by centrifugation using a 10 kDa molecular weight cut-off concentrator (Amicon) and further purified by size exclusion chromatography using a Superdex S200 column (GE Healthcare) pre-equilibrated with buffer E (200 mM KCl, 20 mM Tris-HCl pH 8). The peak fractions were pooled, concentrated, and flash-frozen and stored at –20°C for subsequent experiments.

## Amphipol reconstitution of SiNRAT

For structure determination, the SiNRAT-Nb1 complex was reconstituted into amphipols. To ensure the stability of the proteins during this process, purified SiNRAT and Nb1 were mixed at a 1:2.5 molar ratio for 30 min prior to amphipol reconstitution. The SiNRAT-Nb1 complex and amphipol PMAL-C8 were mixed at a mass ratio of 1:5 and gently stirred overnight at 4°C. DDM was removed by the addition of a 100-fold excess (wt/wt) of biobeads SM-2 (Bio-Rad) compared to protein for 4 hr at 4°C under gentle agitation. The biobeads were removed by filtration and the protein-amphipol mixture was

concentrated before purification by size exclusion chromatography using a Superdex S200 column (GE Healthcare) pre-equilibrated with buffer SEC2 (200 mM NaCl, 20 mM HEPES pH 7). The protein peak fractions were pooled and concentrated to 2.7 mg/ml for cryo-EM grid preparation.

## Cryo-EM sample preparation and data collection

For sample preparation for cryo-EM, 2.5 µl of the complex were applied to glow-discharged holey carbon grids (Quantifoil R1.2/1.3 Au 200 mesh). Samples were blotted for 2–4 s at 4°C and 100% humidity. The grids were frozen in liquid propane-ethane mix using a Vitrobot Mark IV (Thermo Fisher Scientific). All datasets were collected on a 300 kV Titan Krios (Thermo Fisher Scientific) with a 100 µm objective aperture and using a post-column BioQuantum energy filter with a 20 eV slit and a K3 direct electron detector (Gatan) in super-resolution mode. All datasets were recorded automatically using EPU2.9 (Thermo Fisher) with a defocus ranging from –1 to –2.4 µm, a magnification of 130,000× corresponding to a pixel size of 0.651 Å per pixel (0.3255 Å in super-resolution mode) and an exposure of 1.01 s (36 frames). Two datasets were collected for SiNRAT-Nb1 with respective total doses of 69.739 and 69.63 e$^-$/Å$^2$.

## Cryo-EM data processing

Datasets were processed using Cryosparc v3.2.0 (*Punjani et al., 2017*) following the same processing pipeline (*Figure 2—figure supplement 2*). All movies were subjected to motion correction using patch motion correction with a Fourier crop factor of 2 (pixel size of 0.651 Å/pix). After patch CTF estimation, high-quality micrographs were identified based on relative ice thickness, CTF resolution, and total full frame motion and micrographs not meeting the specified criteria were rejected. For the SiNRAT-Nb1 complex, 2D classes generated from the final map of rXkr9 in complex with Sb1$^{rXkr9}$ (*Straub et al., 2021*) were used for template-driven particle picking on the whole dataset.

The 2D classes generated from particles selected from the entire dataset were extracted using a box size of 360 pix and down-sampled to 180 pixels (pixel size of 1.32 Å/pix). These particles were used for generation of four ab initio classes. Promising ab initio models were selected based on visual inspection and subjected to heterogenous refinement using one of the selected models as 'template' and an obviously bad model as decoy model. After several rounds of heterogenous refinement, the selected particles and models were subjected to non-uniform refinement (input model filtering to 8 Å) followed by local CTF refinement and another round of non-uniform refinement. Finally, the maps were sharpened using the sharpening tool from the Cryosparc package. The quality of the map was evaluated validated using 3DFSC (*Tan et al., 2017*) for FSC validation and local resolution estimation.

## Cryo-EM model building and refinement

Model building was performed in Coot (*Emsley and Cowtan, 2004*). Initially, the structure of ScaDMT (PDB 5M94) was rigidly fitted into the density and served as template for map interpretation. The quality of the map allowed for the unambiguous assignment of residues 54–497. The structure of Nb16 (PDB 5M94) was used to build Nb1. The model was iteratively improved by real space refinement in PHENIX (*Afonine et al., 2018*) maintaining secondary structure constrains throughout. Figures were generated using ChimeraX (*Pettersen et al., 2021*) and Dino (http://www.dino3d.org). Surfaces were generated with MSMS (*Sanner et al., 1996*).

## Acknowledgements

This research was supported by the Swiss National Science Foundation (SNF) through the National Center of Competence in Research TransCure. We thank Dr. Marta Sawicka for input in cryo-EM and help during initial sample characterization. Nanobodies were generated at the Nanobody Service Facility of UZH with the help of Dr. Sasa Stefanic. The assistance of Yvonne Neldner during library generation is acknowledged. The cryo-electron microscope and K3-camera were acquired with support of the Baugarten and Schwyzer-Winiker foundations and a Requip grant of the Swiss National Science Foundation. We thank Simona Sorrentino and the Center for Microscopy and Image Analysis (ZMB) of the University of Zurich for their support and access to the electron microscope. All members of the Dutzler lab are acknowledged for help in various stages of the project.

## Additional information

### Funding

| Funder | Grant reference number | Author |
|---|---|---|
| Schweizerischer Nationalfonds zur Förderung der Wissenschaftlichen Forschung | NCCR TransCure | Raimund Dutzler |

The funders had no role in study design, data collection and interpretation, or the decision to submit the work for publication.

### Author contributions

Karthik Ramanadane, Conceptualization, Data curation, Formal analysis, Investigation, Methodology, Supervision, Validation, Visualization, Writing – original draft, Writing – review and editing, Cloned, expressed and purified proteins, performed transport assays and ITC experiments, prepared samples for cryo-EM, processed cryo-EM data and built models; Márton Liziczai, Conceptualization, Data curation, Formal analysis, Investigation, Methodology, Validation, Visualization, Writing – review and editing, Assisted in biochemical and functional experiments, processed cryo-EM data and built models; Dragana Markovic, Data curation, Formal analysis, Validation, Investigation, Visualization, Methodology, Writing – review and editing, Assisted in biochemical and functional experiments; Monique S Straub, Data curation, Formal analysis, Validation, Investigation, Methodology, Writing – review and editing, Collected cryo-EM data and assisted in cryo-EM data processing; Gian T Rosalen, Data curation, Formal analysis, Validation, Investigation, Methodology, Writing – review and editing, Assisted in nanobody selection; Anto Udovcic, Data curation, Formal analysis, Validation, Investigation, Methodology, Writing – review and editing, Assisted in nanobody selection; Raimund Dutzler, Conceptualization, Supervision, Funding acquisition, Validation, Investigation, Visualization, Writing – original draft, Project administration, Writing – review and editing; Cristina Manatschal, Conceptualization, Data curation, Formal analysis, Supervision, Validation, Investigation, Visualization, Methodology, Writing – original draft, Writing – review and editing, Collected and processed ITC data

### Author ORCIDs

Karthik Ramanadane http://orcid.org/0000-0001-7188-0250
Márton Liziczai http://orcid.org/0000-0002-6673-9209
Monique S Straub http://orcid.org/0000-0002-7721-5048
Raimund Dutzler http://orcid.org/0000-0002-2193-6129
Cristina Manatschal http://orcid.org/0000-0002-4907-7303

### Decision letter and Author response

Decision letter https://doi.org/10.7554/eLife.85641.sa1
Author response https://doi.org/10.7554/eLife.85641.sa2

## Additional files

### Supplementary files

• MDAR checklist

### Data availability

The cryo-EM density map of the SiNRAT-Nb1 complex has been deposited in the Electron Microscopy Data Bank under ID code EMD-17000. The coordinates for the atomic model of the SiNRAT-Nb1 complex refined against the 3.66 Å cryo-EM density have been deposited in the Protein Data Bank under ID code 8ONT. Source data files have been provided for *Figure 1*, *Figure 1—figure supplement 2*, *Figure 2—figure supplement 1*, and *Figure 5*.

The following datasets were generated:

| Author(s) | Year | Dataset title | Dataset URL | Database and Identifier |
|---|---|---|---|---|
| Ramanadane K, Liziczai M, Markovic D, Straub MS, Rosalen GT, Udovcic A, Dutzler R, Manatschal C | 2023 | Structure of Setaria italica NRAT in complex with a nanobody | https://www.ebi.ac. uk/emdb/EMD-17000 | Electron Microscopy Data Bank, EMD-17000 |
| Ramanadane K, Liziczai M, Markovic D, Straub MS, Rosalen GT, Udovcic A, Dutzler R, Manatschal C | 2023 | Structure of Setaria italica NRAT in complex with a nanobody | https://doi.org/10. 2210/pdb8ONT/pdb | Worldwide Protein Data Bank, 10.2210/pdb8ONT/ pdb |

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

# Appendix 1

## Appendix 1—key resources table

| Reagent type (species) or resource | Designation | Source or reference | Identifiers | Additional information |
|---|---|---|---|---|
| Cell line (*Homo sapiens*) | HEK293S GnTI- | ATCC | CRL-3022 | Cells were checked for mycoplasma contamination |
| Cell line (*Homo sapiens*) | HEK293T | ATCC | CRL-3216 | Cells were checked for mycoplasma contamination |
| Chemical compound, drug | 1-Palmitoyl-2-oleoyl-*sn*-glycero-3-phospho-(1'rac-glycerol) (18:1 06:0 POPG) | Avanti Polar Lipids | 840457C | |
| Chemical compound, drug | 1-Palmitoyl-2-oleoyl-*sn*-glycero-3-phosphoethanolamine (18:1 06:0 POPE) | Avanti Polar Lipids | 850757C | |
| Chemical compound, drug | HyClone HyCell TransFx-H medium | Cytiva | SH30939.02 | |
| Chemical compound, drug | FBS | Sigma | F2442-500ML | |
| Chemical compound, drug | L-Glutamine | Thermo Fisher Scientific | 25030081 | |
| Chemical compound, drug | PEI | Fisher Scientific | NC1014320 | |
| Chemical compound, drug | PEI Max | Fisher Scientific | NC1038561 | |
| Chemical compound, drug | DMEM-Glutamax, High Glucose | Sigma | D6429 | |
| Chemical compound, drug | Triton X-100 | Sigma | Cat#T9284 | |
| Chemical compound, drug | Lysozyme | Applichem | Cat#A3711 | |
| Chemical compound, drug | Benzamidine | Sigma | B6506 | |
| Chemical compound, drug | Chloroform | Fluka | 25690 | |
| Chemical compound, drug | Egg PC | Avanti Polar Lipids | 840051C | |
| Chemical compound, drug | DDM | Anatrace | D310S | |
| Chemical compound, drug | Diethyl ether | Sigma | 296082 | |
| Chemical compound, drug | DNase I | AppliChem | A3778 | |
| Chemical compound, drug | Glycerol 99% | Sigma | G7757 | |
| Chemical compound, drug | HCl | Merck Millipore | 1.00319.1000 | |
| Chemical compound, drug | HEPES | Sigma | H3375 | |
| Chemical compound, drug | Imidazole | Roth | X998.4 | |
| Chemical compound, drug | L-(+)-arabinose | Sigma | A3256 | |
| Chemical compound, drug | Leupeptin | AppliChem | A2183 | |
| Chemical compound, drug | Pepstatin | AppliChem | A2205 | |

*Appendix 1 Continued on next page*

*Appendix 1 Continued*

| Reagent type (species) or resource | Designation | Source or reference | Identifiers | Additional information |
|---|---|---|---|---|
| Chemical compound, drug | Valinomycin | Thermo Fisher Scientific | V1644 | |
| Chemical compound, drug | Calcimycin | Thermo Fisher Scientific | A1493 | |
| Chemical compound, drug | Ionomycin | Thermo Fisher Scientific | I24222 | |
| Chemical compound, drug | Fura2 | Thermo Fisher Scientific | F1200 | |
| Chemical compound, drug | Magnesium Green | Thermo Fisher Scientific | M3733 | |
| Chemical compound, drug | ACMA | Thermo Fisher Scientific | A1324 | |
| Chemical compound, drug | CCCP | Sigma | C2759 | |
| Chemical compound, drug | Calcein | Thermo Fisher Scientific | C481 | |
| Chemical compound, drug | Phosphate buffered saline | Sigma | D8537 | |
| Chemical compound, drug | D-Desthiobiotin | Sigma | D1411 | |
| Chemical compound, drug | Strep-Tactin Superflow high capacity resin | IBA | 2-1208-010 | |
| Chemical compound, drug | Kolliphor P188 | Sigma | K4894 | |
| Chemical compound, drug | Penicillin- streptomycin | Sigma | P0781 | |
| Chemical compound, drug | Calcium chloride | Sigma | 223506 | |
| Chemical compound, drug | Manganese chloride | Fluka | 31422 | |
| Chemical compound, drug | Magnesium chloride | Fluka | 63065 | |
| Chemical compound, drug | Aluminum chloride | Sigma | 237078 | |
| Chemical compound, drug | Gallium iodide | Sigma | 399116 | |
| Chemical compound, drug | PMSF | Sigma | P7626 | |
| Chemical compound, drug | Potassium chloride | Sigma | 746346 | |
| Chemical compound, drug | Sodium chloride | Sigma | 71380 | |
| Chemical compound, drug | Terrific broth | Sigma | T9179 | |
| Commercial assay, kit | 4–20% Mini-PROTEAN TGX Precast Protein Gels, 15-well, 15 µl | Bio-Rad Laboratories | 4561096DC | |
| Commercial assay, kit | Amicon Ultra-4 Centrifugal Filters Ultracel 10K, 4 ml | Merck Millipore | UFC801096 | |
| Commercial assay, kit | Amicon Ultra-4 Centrifugal Filters Ultracel 50K, 4 ml | Merck Millipore | UFC805096 | |
| Commercial assay, kit | Amicon Ultra-4 Centrifugal Filters Ultracel 100K, 4 ml | Merck Millipore | UFC810024 | |
| Commercial assay, kit | Biobeads SM-2 adsorbents | Bio-Rad Laboratories | 152-3920 | |

*Appendix 1 Continued on next page*

*Appendix 1 Continued*

| Reagent type (species) or resource | Designation | Source or reference | Identifiers | Additional information |
|---|---|---|---|---|
| Commercial assay, kit | Avestin Extruder kit | Sigma | Cat#Z373400 | |
| Commercial assay, kit | 400 nm polycarbonate filters | Sigma | Cat#Z373435 | |
| Commercial assay, kit | 96-Well black-walled microplates | Thermo Fisher Scientific | Cat#M33089 | |
| Commercial assay, kit | Ni-NTA resin | ABT Agarose Bead Technologies | 6BCL-NTANi-X | |
| Commercial assay, kit | QuantiFoil R1.2/1.3 Au 200 mesh | Electron Microscopy Sciences | Q2100AR1.3 | |
| Commercial assay, kit | Superdex 200 10/300 GL | Cytiva | 17517501 | |
| Commercial assay, kit | Superdex 200 Increase 3.2/300 | Cytiva | 28990946 | |
| Commercial assay, kit | Superdex 200 Increase 5/150 GL | Cytiva | 28990945 | |
| Commercial assay, kit | Superdex 75 10/300 GL | Cytiva | 17517401 | |
| Other | BioQuantum Energy Filter | Gatan | NA | Electron microscope component |
| Other | HPL6 | Maximator | NA | Cell lyser |
| Other | K3 Direct Detector | Gatan | NA | Electron microscope component |
| Other | Titan Krios G3i | Thermo Fisher Scientific | NA | Electron microscope |
| Other | TECAN M1000 Infinite | TECAN | NA | Used for in vitro transport assays |
| Other | Vitrobot Mark IV | Thermo Fisher Scientific | NA | Used for freezing cryo-EM grids |
| Recombinant DNA reagent | gDNA *Eremococcus coleocola* | DSMZ | 15696 | |
| Recombinant DNA reagent | Bacterial expression vector with C-terminal 3C cleavage site, GFP-tag and His-tag | Dutzler laboratory | NA | |
| Recombinant DNA reagent | Mammalian expression vector with C terminal 3C cleavage site, GFP, Myc, and SBP tag | Dutzler laboratory | NA | |
| Recombinant DNA reagent | Bacterial expression vector with N terminal pelB sequence, His-tag, MBP, 3C cleavage site | Dutzler laboratory | NA | |
| Peptide, recombinant protein | HRV 3C protease | Expressed (pET_3C) and purified in Dutzler laboratory | NA | |
| Software, algorithm | 3DFSC | *Tan et al., 2017* | https://3dfsc.salk.edu/ | |
| Software, algorithm | ASTRA7.2 | Wyatt Technology | https://www.wyatt.com/products/software/astra.html | RRID:SCR_016255 |
| Software, algorithm | Chimera v.1.15 | *Pettersen et al., 2004* | https://www.cgl.ucsf.edu/chimera/ | RRID:SCR_004097 |
| Software, algorithm | ChimeraX v.1.1.1 | *Pettersen et al., 2021* | https://www.rbvi.ucsf.edu/chimerax/ | RRID:SCR_015872 |
| Software, algorithm | Coot v.0.9.4 | *Emsley and Cowtan, 2004* | https://www2.mrc-lmb.cam.ac.uk/personal/pemsley/coot/ | RRID:SCR_014222 |
| Software, algorithm | cryoSPARC v.3.0.1/v.3.2.0 | Structura Biotechnology Inc | https://cryosparc.com/ | RRID:SCR_016501 |
| Software, algorithm | DINO | | http://www.dino3d.org | RRID:SCR_013497 |
| Software, algorithm | EPU2.9 | Thermo Fisher Scientific | NA | |

*Appendix 1 Continued on next page*

*Appendix 1 Continued*

| Reagent type (species) or resource | Designation | Source or reference | Identifiers | Additional information |
|---|---|---|---|---|
| Software, algorithm | Phenix | *Liebschner et al., 2019* | https://www.phenix-online.org/ | RRID:SCR_014224 |
| Software, algorithm | RELION 3.0.7 | *Zivanov et al., 2018* | https://www3.mrc-lmb.cam.ac.uk/relion/ | RRID:SCR_016274 |
| Software, algorithm | WEBMAXC calculator | *Bers et al., 2010* | https://somapp.ucdmc.ucdavis.edu/pharmacology/bers/maxchelator/webmaxc/webmaxcS.htm | RRID:SCR_018807 |
| Software, algorithm | JALVIEW | *Waterhouse et al., 2009* | | |
| Software, algorithm | Muscle | *Edgar, 2004* | | |
| Strain, strain background (*Escherichia coli*) | MC1061 | Thermo Fisher Scientific | C66303 | |

