## [Editor Report]

This study uses cryo-EM and functional assays to characterize a new subtype of transporters from the NRAMP family that protect plants against aluminum toxicity. Evidence of divalent metal ion transport and the structure (obtained without added metal ions) are convincing. Although technical challenges inherent to the structural and functional characterization of eukaryotic membrane proteins prevent a complete molecular description of Al^3+^ binding and transport, the work nonetheless provides new and valuable insight into the diversity within the NRAMP superfamily of transporters, and will be of interest to structural biologists and biophysicists studying NRAMP transporters.

---

## [Decision Letter]

**Decision letter after peer review:**

Thank you for submitting your article "Structural and functional properties of a plant NRAMP related aluminum 1 transporter" for consideration by *eLife*. Your article has been reviewed by 3 peer reviewers, including Randy B Stockbridge as the Reviewing Editor and Reviewer #1, and the evaluation has been overseen by Kenton Swartz as the Senior Editor. The following individual involved in the review of your submission has agreed to reveal their identity: Motoyuki Hattori (Reviewer #2).

Essential revisions:

1. Figure 3 —figure supplement 1 is missing and should be included.

2. The reviewers ask that the authors provide additional discussion on the following points:

a. Could the authors describe whether attempts were made to collect a cryo-EM dataset with divalent or trivalent metals (Mn^2+^, ca^2+^, Mg^2+^, Al3+, or Ga3+) present, and what the results were?

b. The result that passive metal transport (uptake) is protective against high environmental aluminum is counterintuitive. However, the proposal (suggested by divalent metal transport) is in line with the model from the previous literature (the Xia et al. reference cited in the manuscript) in which these proteins prevent aluminum disruption of the cell wall by transporting it to the cytoplasm where it can be more effectively sequestered. Could the authors add some text to the discussion explaining this uptake-followed-by-detoxification model, which the current results seem to support?

c. What, if anything, is known about the pH dependence of this transporter? Presumably lower pH values (i.e. pH 5.5) are more physiologically relevant and would favor Al3+, simplifying the binding analysis. Did the authors attempt experiments in this range?

d. The authors justify a lack of Al3+ transport data by explaining how this ion is disruptive to membranes. If this is true, it raises a fundamental question how is this transporter able to work in a physiological environment and how are cells membranes able to withstand physiological levels of Al3+ when reconstituted proteoliposomes are not?

e. Based on the cryo-EM structure of the SiNRAT-Nb1 complex, please provide a possible explanation for the partial inhibition of Mg^2+^ transport by Nb1.

3. The reviewers had some questions about some of the structural interpretations, which should be addressed in the text:

a. The text states that a shorter side chain for Ser403 in SiNRAT – compared to a Gln in DraNRAMP – would make an impact on metal binding. However, the earlier statement that the backbone carbonyl is responsible for the relevant interaction presents a contradiction.

b. Thr241 is part of a signature sequence that distinguishes the NRAT clade and the importance of this residue is confirmed by mutagenesis. However, the coordinates provides show a rotamer with the oxygen atom pointing away from the ion binding site, which seems inconsistent with its presumed role.

c. There are elongated EM densities at the cavity composed of alpha10-alpha12, which may be derived from lipid molecules. Could the authors please add a comment on this putative lipid binding?

d. The general description of the SiNRAT structure includes a comparison with EcoDMT, but this is not shown in the figures. Such a comparison would be useful to illustrate the disposition of alpha11 and alpha12. RMSD values are cited, but it is not clear if this calculation applies to Calpha atoms only. Finally, Figure 3 shows superposition with 6C3I but the text refers to 8E60, which is not yet released by the PDB.

4. The clarity of figures 3 and 4 should be improved according to the following suggestions:

a. The structural nature of the occluded state is not well depicted in the figures (perhaps this was shown in the missing Figure 3 supplement). The authors state that the substrate-binding pocket is larger for SiNRAT, but the size is not depicted and it is not clear if this conclusion is based on a quantitative measurement or just a visual inspection of the site.

b. In Figures 3c, 4b, and 4c, the inset is not well-distinguished from the main part of the figure. This could be solved easily by adding a dashed box around the inset in order to better distinguish it from the main part of the figure.

5. In the first paragraph of the Discussion, the authors say that they have shown the transport of Mn, Ca, and Mg with μM Km. This is not accurate as they have only shown the transport of Mn and have not measured Km for Ca and Mg. This raises the question of why the authors do not quantitate rates for Ca and Mg-based on the data shown in Figure 1 E and F, which should provide at least a rough estimate of Km. In addition, please provide the error value for the KM of Mn^2+^ (Page 6, lines 96-98).

6. Page 54, Figure 6. Please provide a figure for the sequence alignment on the metal binding sites in alpha1 and alpha6 using several NRAMP, NRMT, and NRAT proteins.

7. The discussion of proton-coupling (pages 9-10) is too vague and speculative without experimental evidence to support these proposals. This section should be shortened/removed since the molecular basis for the lack of proton coupling is still unknown.

*Reviewer #1 (Recommendations for the authors):*

1. Enthusiasm for the story is dampened somewhat by the uncertainty of the density in the binding site. I imagine the specific sidechains would rearrange considerably if the density is water vs. a Na^+^ vs a residual divalent. Did the authors try to collect a cryo-EM dataset with transition metals (Mn^2+^, ca^2+^, Mg^2+^, or Ga3+) present?

2. Lack of information about the structure or function at lower pH values seems like an important gap in understanding the system. From a physiological perspective, Al3+ is the predominant species in low-pH environments, and aluminum remediation is especially important in low-pH environments, so it's possible that the conformation and transport activity at pH 5 or 5.5 might be more physiologically relevant than at pH 7 or 7.5. (Lower pH would also likely simplify the ITC experiments). From a biophysical perspective, given the density of the negative charge in the aqueous chamber, it is plausible that Asps are protonated, even at pH 7 or 7.5. The role of pH in substrate binding might therefore be mechanistically important as well.

3. For a transporter person, the suggestion that a passive transporter could be useful for providing resistance against environmental aluminum is confusing. I anticipated that aluminum export would be the desired transport function. However, it seems like this counterintuitive result is in line with the model from the previous literature (the Xia et al. reference cited in the ms) in which these proteins prevent aluminum disruption of the cell wall by transporting it to the cytoplasm where it can be more effectively sequestered. To make this more intuitive to people who think about ion gradients, could the authors add some text to the discussion explaining this uptake followed by the detoxification model, which the current results seem to support? Could the authors also comment on whether NRATs are broadly distributed in plants, or whether they are only found in a subset?

4. There are a number of figures with insets where I was confused about whether I was looking at a component of the main figure or an inset (Figure 3c, 4b, 4c). This could be solved easily by adding a dashed box or something around the inset in order to better distinguish it from the main part of the figure.

5. Figure 3 —figure supplement 1 is missing.

*Reviewer #2 (Recommendations for the authors):*

While I highly recommend this work for publication, there are a few concerns to be addressed before publication.

1. Page 6, Lines 96-98. "Transport is strongly dependent on the Mn^2+^ concentration and saturates with an apparent KM of 13 µM reflecting the binding of the transported substrate to a saturable site (Figure 1A, B)."

Please show the error value for KM.

2. Page 8, Lines 148-151. "The procedure allowed the identification of the nanobody Nb1SiNRAT (short Nb1) as a promising interaction partner that does not dissociate during size exclusion chromatography and which partly inhibits Mn^2+^ transport when added to the outside of liposomes (Figure 2—figure supplement 1, Table 2)."

Based on the cryo-EM structure of the SiNRAT-Nb1 complex, please provide a possible explanation for the partial inhibition of Mg^2+^ transport by Nb1.

3. Page 54, Figure 6.

Please provide a figure for the sequence alignment on the metal binding sites in alpha1 and alpha6 using several NRAMP, NRMT, and NRAT proteins.

4. Coordinates and map files

There are elongated EM densities at the cavity composed of alpha10-alpha12, which may be derived from lipid molecules. If the authors agree, please provide a possible mechanistic implication of these putative lipid binding.

*Reviewer #3 (Recommendations for the authors):*

Figure 3. Suppl 1 appears to be missing.

The authors justify a lack of Al3+ transport data by explaining how this ion is disruptive to membranes. If this is true, it raises a fundamental question how is this transporter able to work in a physiological environment and how are cells membranes able to withstand physiological levels of Al3+ when reconstituted proteoliposomes are not?

The general description of the SiNRAT structure includes a comparison with EcoDMT, but this is not shown in the figures. Such a comparison would be useful to illustrate the disposition of alpha11 and alpha12. RMSD values are cited, but it is not clear if this calculation applies to Calpha atoms only. Finally, Figure 3 shows superposition with 6C3I but the text refers to 8E60, which is not yet released by the PDB.

The structural nature of the occluded state is not well depicted in the figures (perhaps this was shown in the missing Figure 3 supplement). The authors state that the substrate-binding pocket is larger for SiNRAT, but the size is not depicted and it is not clear if this conclusion is based on a quantitative measurement or just a visual inspection of the site.

The text states that a shorter side chain for Ser403 in SiNRAT – compared to a Gln in DraNRAMP – would make an impact on metal binding. However, the earlier statement that the backbone carbonyl is responsible for the relevant interaction presents a contradiction.

Thr241 is part of a signature sequence that distinguishes the NRAT clade and the importance of this residue is confirmed by mutagenesis. However, the coordinates provides show a rotamer with the oxygen atom pointing away from the ion binding site, which seems inconsistent with its presumed role.

The discussion of proton coupling on pg 9-10 is quite vague and speculative and not particularly convincing.

In the first paragraph of the Discussion, the authors say that they have shown the transport of Mn, Ca, and Mg with μM Km. This is not accurate as they have only shown the transport of Mn and have not measured Km for Ca and Mg. This raises the question of why the authors do not quantitate rates for Ca and Mg-based on the data shown in Figure 1 E and F, which should provide at least a rough estimate of Km.

The authors point out that Asp140 provides an extra negative charge at the ion binding site, which could help balance the extra charge associated with trivalent ions (vs. divalent). It is therefore curious that they did not analyze the corresponding mutant.

Finally, the take-home message of this paper is a bit diffuse. A major new finding seems to be that Ser68 governs the alleged specificity for trivalent cations and the authors might be well advised to use this as a focal point in their presentation.

---

## [Author Response]

Essential revisions:1. Figure 3 —figure supplement 1 is missing and should be included.

We have now included Figure 3—figure supplement 1 and added two panels to Figure 2—figure supplement 4.

2. The reviewers ask that the authors provide additional discussion on the following points:a. Could the authors describe whether attempts were made to collect a cryo-EM dataset with divalent or trivalent metals (Mn^2+^, ca^2+^, Mg^2+^, Al3+, or Ga3+) present, and what the results were?

No attempts were made with regard to the structure determination of SiNRAT in presence of transported ions. As mentioned in the text, the structure determination of SiNRAT was challenging and has required the binding of a nanobody and the reconstitution of the protein into amphipols, since previous efforts to determine the structure of the SiNRAT/nanobody complex in detergents did not permit reconstruction at high resolution. The clearly most instructive structure would be in complex with its substrate Al^3+^. However, due to the small mass of Al^3+^, it would likely not be resolved at the obtained resolution of the data, as we experienced in structures of other transport proteins determined in our lab. Moreover, even in complex structures where we were able to resolve bound ions, these had to be supplemented at high concentration (several times above their respective K_M_). Due to the low solubility of trivalent ions at neutral pH, this would have required a decrease of the pH, which leads to the dissociation of the nanobody.

We have added a sentence to the results (lines 163-165):

“Due to the challenges associated with the amphipol reconstitution process, the sample used for imaging did not contain transported ions. In case of trivalent ions, this would have required a lowering of the pH to increase their solubility, which was incompatible with nanobody binding.”

b. The result that passive metal transport (uptake) is protective against high environmental aluminum is counterintuitive. However, the proposal (suggested by divalent metal transport) is in line with the model from the previous literature (the Xia et al. reference cited in the manuscript) in which these proteins prevent aluminum disruption of the cell wall by transporting it to the cytoplasm where it can be more effectively sequestered. Could the authors add some text to the discussion explaining this uptake-followed-by-detoxification model, which the current results seem to support?

We have added a sentence to the discussion, but also want to emphasize that our work does not aim to clarify the mechanism of Al^3+^ detoxification in plants.

Line 361-364:

“Despite these ambiguities, our findings are in general agreement with a previous proposal that the NRAMP-mediated uptake of Al^3+^ into cells and its subsequent detoxification by complexation with organic anions and import into the vacuolar compartment would decrease its ability to interfere with cell wall integrity.”

c. What, if anything, is known about the pH dependence of this transporter? Presumably lower pH values (i.e. pH 5.5) are more physiologically relevant and would favor Al3+, simplifying the binding analysis. Did the authors attempt experiments in this range?

We noticed a stabilization of SiNRAT at lower pH but refrained from transport experiments under such conditions as the pH alters the fluorescence properties of the employed metal-sensitive fluorophores. We also want to emphasize that H^+^-coupled transition metal ion transporters of the SLC11/NRAMP family, assayed under the same conditions, show robust transport properties, despite the fact that, in a cellular environment, their extracellular side would face a lower pH. Similarly, in our assays with SiNRAT, we observe pronounced activity for the investigated divalent ions already at neutral pH. We thus suppose that the lowering of the pH would exert its predominant effect on the solubility of the trivalent ions Al^3+^ and Ga^3+^, which we were not able to assay directly and where we noticed an effect of the pH decrease on trivalent ion-mediated leakiness of proteoliposomes.

With respect to ITC, we want to emphasize the technical challenges associated with the use of this technique for poorly expressed eukaryotic membrane proteins. In this method, the measured signal depends on the enthalpic contribution of the binding event for detection. In case of a lower pH, we noticed a severe decrease of the exchanged heat, which presumably originates from the lower binding enthalpy. We thus clearly acknowledge the limited insight of our assays for trivalent ion binding and transport in our manuscript.

Line 96-99:

“Although, in a physiological context, SiNRAT presumably operates at mildly acidic extracellular conditions (i.e., at a pH < 5.5), all transport experiments were carried out at neutral pH since this resulted in maximum proteoliposome stability and highest sensitivity of the employed fluorophores.”

Line 121-124:

“Thus, despite the low solubility of Ga^3+^ under the assayed conditions (Yu and Liao, 2011), which obscures a definitive assignment of its concentration dependence, our results suggest a specific effect of the ion on SiNRAT-mediated transport.”

Line 127-133:

“However, at the neutral pH of the sample, the trivalent ion is at equilibrium with its hydroxyl complexes, which complicates data analysis. The thermograms were thus best fitted to a sum of two binding constants, both in the low μM range (Figure 1I). Although precluding a definitive mechanistic interpretation, we use the ITC results of Al^3+^ binding as a qualitative assay of its interaction with SiNRAT and show later that the event is directly associated with its binding to the site that is relevant for ion transport.”

Line 313-317:

“Although the direct assay of transport of trivalent ions such as Ga^3+^ and Al^3+^ was prohibited by technical limitations, we have demonstrated the interaction of SiNRAT with both ions, which defines them as plausible substrates (Figure 1G-I). However, due to the poor solubility of these trivalent metal ions under the assay conditions, these results are tentative and should be taken with caution.”

d. The authors justify a lack of Al3+ transport data by explaining how this ion is disruptive to membranes. If this is true, it raises a fundamental question how is this transporter able to work in a physiological environment and how are cells membranes able to withstand physiological levels of Al3+ when reconstituted proteoliposomes are not?

We have noticed a concentration-dependent non-specific leak in experiments assaying the competition of Mn^2+^ by Al^3+^ already at low µM Al^3+^ concentrations. The same effect was also observed in mock-reconstituted control liposomes, which has prohibited any competition experiments with Al^3+^. This corroborates the described properties of Al^3+^ to interfere with membrane integrity (presumably by forming tight complexes with the phosphate groups). However, it should be emphasized that the detailed stability properties of a liposome and the membrane of a plant cell presumably differ and are influenced by the exact lipid composition and the properties of the surrounding cell-wall.

e. Based on the cryo-EM structure of the SiNRAT-Nb1 complex, please provide a possible explanation for the partial inhibition of Mg^2+^ transport by Nb1.

We assume that the partial inhibition mainly reflects the mixed inside-out and outside-out orientation of SiNRAT in proteoliposomes. Based on its structure, we expect the bound nanobody to lock the observed conformation, thereby inhibiting transport. However, when applied to the outside, the nanobody only recognizes its epitope in proteins residing in an outside-out orientation, and its application would thus only affect the function of this fraction.

We have added the following sentence line 156-158:

“The incomplete inhibition of transport by Nb1 presumably reflects the mixed distribution of reconstituted SiNRAT residing in either inside-out or outside-out orientation, where the epitopes recognized by the nanobody are only accessible in one orientation.”

3. The reviewers had some questions about some of the structural interpretations, which should be addressed in the text:a. The text states that a shorter side chain for Ser403 in SiNRAT – compared to a Gln in DraNRAMP – would make an impact on metal binding. However, the earlier statement that the backbone carbonyl is responsible for the relevant interaction presents a contradiction.

This was a mistake in our original manuscript. It is the side-chain of Gln 378 in DraNRAMP, not its backbone, which interacts with the bound ion in DraNRAMP (presumably via a bound water as shown in Figure 4C). The shortened sidechain of Ser 403 does not contact the aqueous cavity and we thus suppose that it would not be involved in either direct or indirect ion interactions.

We have introduced the correction line 228-229:

“Finally, a water-mediated interaction with the side chain of Gln 378 on α10 completes the octahedral coordination of the ion (Figure 4C).”

b. Thr241 is part of a signature sequence that distinguishes the NRAT clade and the importance of this residue is confirmed by mutagenesis. However, the coordinates provides show a rotamer with the oxygen atom pointing away from the ion binding site, which seems inconsistent with its presumed role.

The resolution of the map is insufficient to define the correct rotamer of the threonine with certainty. We have changed it such that the hydroxyl faces the aqueous cavity. However, due to the uncertain definitive position of the bound ion, we do not know whether it would engage in direct or water-mediated interactions. We also cannot exclude a local reorientation upon metal binding.

c. There are elongated EM densities at the cavity composed of alpha10-alpha12, which may be derived from lipid molecules. Could the authors please add a comment on this putative lipid binding?

We have now interpreted this density as a phospholipid, which was included in our refined structure. However, the location of the lipid is ambiguous, with a position of the headgroup in the assumed core of the membrane providing a slightly better fit. Such an orientation would not be assumed in a membrane environment. We also do not think that this lipid would be of direct functional relevance.

Legend to Figure 2 line 809-810:

“A, B Proteins are shown in unique colors, the density of a bound lipid with uncertain orientation in A is shown in grey.”

d. The general description of the SiNRAT structure includes a comparison with EcoDMT, but this is not shown in the figures. Such a comparison would be useful to illustrate the disposition of alpha11 and alpha12. RMSD values are cited, but it is not clear if this calculation applies to Calpha atoms only. Finally, Figure 3 shows superposition with 6C3I but the text refers to 8E60, which is not yet released by the PDB.

The superposition of SiNRAT and EcoDMT is now provided in Figure 2—figure supplement 4B. The RMSD was calculated based on Cα-positions. It is at that stage not clear in how much the observed α11-12 conformation would be influenced by the nanobody as there are currently no other structures of family members containing 12 α-helices available, which reside in an occluded conformation. However, since the respective α-helices 11 in the occluded conformations of SiNRAT at EcoNRAMP are close (Figure 3B), we do not expect a pronounced influence of the bound nanobody on helix conformation. The PDBID 8E60 shows an improved structure of 6C3I, with updated ion coordination. The coordinates were generously provided by Prof. Rachelle Gaudet prior to release.

We have introduced the following changes line 176-179:

“The protein contains a total of twelve membrane-inserted helices akin to other eukaryotic family members and the prokaryotic EcoDMT (PDBID: 5M87) (Ehrnstorfer et al., 2017), whose Cα-positions superimpose with an RMSD of 2.66 Å (Figures 2B and 3A, Figure 2—figure supplement 4B).”

4. The clarity of figures 3 and 4 should be improved according to the following suggestions:a. The structural nature of the occluded state is not well depicted in the figures (perhaps this was shown in the missing Figure 3 supplement). The authors state that the substrate-binding pocket is larger for SiNRAT, but the size is not depicted and it is not clear if this conclusion is based on a quantitative measurement or just a visual inspection of the site.

We have now tried to better document this property in Figure 3—figure supplement 1 D, where a section through the molecular surface (grey) and the surface of the occluded cavity (magenta) are displayed. We have also quantified the volume of the binding site and found it three times larger in SiNRAT (6,071 Å^3^) compared to DraNRAMP (2,404 Å^3^). It was also emphasized that in DraNRAMP, the bound ion is only in contact with the aqueous cavity on one side where a water molecule presumably occupies one of the coordination sites, whereas in SiNRAT the pocket appears to be located in the center of the presumed anion binding site, potentially providing room for additional solvent-mediated interactions. However, since the described structure was obtained in absence of transported ions, this remains at this stage hypothetical.

We have rephrased the text: Line 223-225

“In its binding site, the interacting transition metal ion in DraNRAMP^occ^ is predominantly enclosed by protein residues except for one contact with a small pocket harboring trapped water molecules.”

b. In Figures 3c, 4b, and 4c, the inset is not well-distinguished from the main part of the figure. This could be solved easily by adding a dashed box around the inset in order to better distinguish it from the main part of the figure.

We have added a grey box surrounding the insets in the indicated figure panels.

5. In the first paragraph of the Discussion, the authors say that they have shown the transport of Mn, Ca, and Mg with μM Km. This is not accurate as they have only shown the transport of Mn and have not measured Km for Ca and Mg. This raises the question of why the authors do not quantitate rates for Ca and Mg-based on the data shown in Figure 1 E and F, which should provide at least a rough estimate of Km. In addition, please provide the error value for the KM of Mn^2+^ (Page 6, lines 96-98).

For Mg^2+^ and ca^2+^ this claim was made based on the observed concentration dependence of transport where we find a large transition in the kinetics to occur in the µM range. We have also fitted the data and confirmed this behavior. However, since the signal is in both cases comparably weak, we refrained from showing a fit in the main figure and instead added them as panels F and H of Figure 1—figure supplement 2. Additionally, it should be emphasized that, due to the comparably low affinity of the fluorophore Magnesium Green for Mg^2+^, it is unclear whether the observed concentration dependence reflects the saturation of the binding site of the transporter, which we have stated in the legend to the figure panel. We have provided an error for the K_M_ of Mn^2+^, ca^2+^ and Mg^2+^ in our revised manuscript.

Line 789-793:

“Mg^2+^ concentration dependence of transport. Initial velocities were derived from individual traces of experiments displayed in Figure 1F the solid line shows the fit to a Michaelis–Menten equation with an apparent *K*_m_ of 415 ± 218 μM (with the error based on a 95% confidence interval). Due to the low affinity of Magnesium Green for Mg^2+^, it is uncertain whether the measurements reflect the saturation of the ion binding site in SiNRAT.”

6. Page 54, Figure 6. Please provide a figure for the sequence alignment on the metal binding sites in alpha1 and alpha6 using several NRAMP, NRMT, and NRAT proteins.

We have included such alignment as Figure 1—figure supplement 1.

7. The discussion of proton-coupling (pages 9-10) is too vague and speculative without experimental evidence to support these proposals. This section should be shortened/removed since the molecular basis for the lack of proton coupling is still unknown.

We have shortened the paragraph concerning residues with potential relevance for proton transport in the results and removed this point in the discussion. However, there is ample of literature on proton coupling in classical transition metal ion transporters of the SLC11/NRAMP family, which points towards an involvement of the described residues. Similar to our previous study on the NRAMP-related Mg^2+^ transporter EleNRMT, our data did not provide any evidence for H^+^ coupling in SiNRAT. In our previous manuscript, we have ascribed this to the altered properties of the presumed proton-pathway, which included the replacement of the conserved His on α6 and alterations of the polar region between the proximal acidic and basic residues on α3 and α9. Whereas former is also replaced in SiNRAT, most of the latter residues are found in its structure at equivalent positions. While we do not want to draw definitive conclusions, we still think that these structural features deserve to be described, particularly in relation to the manuscript our work refers to.

We have moved the panels showing the described regions to a figure supplement (Figure 4—figure supplement 1) and added the equivalent region of EleNRMT for comparison.

Line 253-267:

“Although the detailed mechanism of H^+^ coupling in the SLC11/NRAMP family is still a matter of debate, there are remarkable structural differences between classical H^+^-coupled metal ion transporters, SiNRAT, and the Mg^2+^ transporter EleNRMT, (Figure 4—figure supplement 1). In NRAMPs, H^+^ transport was associated with acidic and basic residues located on the α-helices 1, 3, 6, and 9, most of which are replaced by hydrophobic sidechains in NRMTs (Figure 1—figure supplement 1, Figure 4—figure supplement 1B, C). In contrast, many of these residues are conserved in NRATs, which are evolutionally closer to NRAMPs (Figure 1—figure supplement 1, Figure 4—figure supplement 1A). A pronounced difference between NRATs and NRAMPs concerns the substitution of a conserved histidine on α6b (i.e., His 232 in DraNRAMP with a tyrosine (Tyr 243 in SiNRAT)), whereas the equivalent position in NRMTs is occupied by a tryptophane (*i.e*, W229 in EleNRMT) (Figure 1—figure supplement 1, Figure 4—figure supplement 1). However, the exact relation of the described structural features to the observed transport mechanism is still unclear and requires further investigation.”

Reviewer #1 (Recommendations for the authors):1. Enthusiasm for the story is dampened somewhat by the uncertainty of the density in the binding site. I imagine the specific sidechains would rearrange considerably if the density is water vs. a Na^+^ vs a residual divalent. Did the authors try to collect a cryo-EM dataset with transition metals (Mn^2+^, ca^2+^, Mg^2+^, or Ga3+) present?

See response 2. We did not investigate complexes of transported ions for the described reasons. Although we cannot exclude the possibility of local sidechain rearrangements upon ion binding this does not necessarily have to be pronounced. E. g., we did not find an obvious rearrangement (at the resolution of the data) in our initial structure of the prototypic SLC11 transporter ScaDMT.

2. Lack of information about the structure or function at lower pH values seems like an important gap in understanding the system. From a physiological perspective, Al3+ is the predominant species in low-pH environments, and aluminum remediation is especially important in low-pH environments, so it's possible that the conformation and transport activity at pH 5 or 5.5 might be more physiologically relevant than at pH 7 or 7.5. (Lower pH would also likely simplify the ITC experiments). From a biophysical perspective, given the density of the negative charge in the aqueous chamber, it is plausible that Asps are protonated, even at pH 7 or 7.5. The role of pH in substrate binding might therefore be mechanistically important as well.

See response 2c. We have explained why we refrained from transport experiments at low pH above. In any case, in a cellular environment, the low pH would only be experienced on one side (outside) and robust transport properties were observed at neutral pH for all coupled and uncoupled transporters of the SLC11/NRAMP family. We thus do not necessarily expect a qualitative different behavior under such conditions. ITC experiments at low pH have led to a decrease of the signal, probably as a consequence of the decreased binding enthalpy at these conditions. Although we cannot exclude a change in the protonation of any of the acidic residues in the binding site, their pK_a_ would have to be perturbed to be protonated at pH 5.5 and their negative charge would presumably be further stabilized by the bound cation.

3. For a transporter person, the suggestion that a passive transporter could be useful for providing resistance against environmental aluminum is confusing. I anticipated that aluminum export would be the desired transport function. However, it seems like this counterintuitive result is in line with the model from the previous literature (the Xia et al. reference cited in the ms) in which these proteins prevent aluminum disruption of the cell wall by transporting it to the cytoplasm where it can be more effectively sequestered. To make this more intuitive to people who think about ion gradients, could the authors add some text to the discussion explaining this uptake followed by the detoxification model, which the current results seem to support? Could the authors also comment on whether NRATs are broadly distributed in plants, or whether they are only found in a subset?

We have mentioned that the abundance of NRATs is scarce (we essentially only identified 7 orthologues). We have explicitly mentioned the work by Xia but prefer to refrain from a discussion of potential physiological mechanisms, as this is not topic of our study. In any case would a negative membrane potential favor the uptake of the strongly positively charged cargo.

4. There are a number of figures with insets where I was confused about whether I was looking at a component of the main figure or an inset (Figure 3c, 4b, 4c). This could be solved easily by adding a dashed box or something around the inset in order to better distinguish it from the main part of the figure.

We have corrected that.

5. Figure 3 —figure supplement 1 is missing.

We have inserted the figure.

Reviewer #2 (Recommendations for the authors):While I highly recommend this work for publication, there are a few concerns to be addressed before publication.1. Page 6, Lines 96-98. "Transport is strongly dependent on the Mn^2+^ concentration and saturates with an apparent KM of 13 µM reflecting the binding of the transported substrate to a saturable site (Figure 1A, B)."Please show the error value for KM.

We have now included the error for the KM values.

2. Page 8, Lines 148-151. "The procedure allowed the identification of the nanobody Nb1SiNRAT (short Nb1) as a promising interaction partner that does not dissociate during size exclusion chromatography and which partly inhibits Mn^2+^ transport when added to the outside of liposomes (Figure 2—figure supplement 1, Table 2)."Based on the cryo-EM structure of the SiNRAT-Nb1 complex, please provide a possible explanation for the partial inhibition of Mg^2+^ transport by Nb1.

See answer to point 2e above.

3. Page 54, Figure 6.Please provide a figure for the sequence alignment on the metal binding sites in alpha1 and alpha6 using several NRAMP, NRMT, and NRAT proteins.

This was added as Figure 1—figure supplement 1.

4. Coordinates and map filesThere are elongated EM densities at the cavity composed of alpha10-alpha12, which may be derived from lipid molecules. If the authors agree, please provide a possible mechanistic implication of these putative lipid binding.

*­*See response to point 3c above.

Reviewer #3 (Recommendations for the authors):Figure 3. Suppl 1 appears to be missing.

We have added the figure.

The authors justify a lack of Al3+ transport data by explaining how this ion is disruptive to membranes. If this is true, it raises a fundamental question how is this transporter able to work in a physiological environment and how are cells membranes able to withstand physiological levels of Al3+ when reconstituted proteoliposomes are not?

See response to point 2d above.

The general description of the SiNRAT structure includes a comparison with EcoDMT, but this is not shown in the figures. Such a comparison would be useful to illustrate the disposition of alpha11 and alpha12. RMSD values are cited, but it is not clear if this calculation applies to Calpha atoms only. Finally, Figure 3 shows superposition with 6C3I but the text refers to 8E60, which is not yet released by the PDB.

See response to point 3d above.

The structural nature of the occluded state is not well depicted in the figures (perhaps this was shown in the missing Figure 3 supplement). The authors state that the substrate-binding pocket is larger for SiNRAT, but the size is not depicted and it is not clear if this conclusion is based on a quantitative measurement or just a visual inspection of the site.

See response to point 4a above.

The text states that a shorter side chain for Ser403 in SiNRAT – compared to a Gln in DraNRAMP – would make an impact on metal binding. However, the earlier statement that the backbone carbonyl is responsible for the relevant interaction presents a contradiction.

See response to point 3a above.

Thr241 is part of a signature sequence that distinguishes the NRAT clade and the importance of this residue is confirmed by mutagenesis. However, the coordinates provides show a rotamer with the oxygen atom pointing away from the ion binding site, which seems inconsistent with its presumed role.

See response to point 3b above.

The discussion of proton coupling on pg 9-10 is quite vague and speculative and not particularly convincing.

See response to point 7 above.

In the first paragraph of the Discussion, the authors say that they have shown the transport of Mn, Ca, and Mg with μM Km. This is not accurate as they have only shown the transport of Mn and have not measured Km for Ca and Mg. This raises the question of why the authors do not quantitate rates for Ca and Mg-based on the data shown in Figure 1 E and F, which should provide at least a rough estimate of Km.

See response to point 5 above.

The authors point out that Asp140 provides an extra negative charge at the ion binding site, which could help balance the extra charge associated with trivalent ions (vs. divalent). It is therefore curious that they did not analyze the corresponding mutant.

We expect Asp 140 to be important for the interaction with trivalent but presumably not with divalent cations. Since our assays for trivalent ions were limited, we plan to characterize this position in detail in a follow-up study.

Finally, the take-home message of this paper is a bit diffuse. A major new finding seems to be that Ser68 governs the alleged specificity for trivalent cations and the authors might be well advised to use this as a focal point in their presentation.

We agree that Ser 68 appears to have a role in the interaction with trivalent cations. However, like the presumed importance of Asp 140, its definitive role in the selection of trivalent ions will have to be investigated in greater detail in a follow-up study.